# CONTEXT-FREE RECOGNITION WITH TRANSFORMERS

## ABSTRACT

Transformers excel on tasks that process well-formed inputs according to some grammar, such as natural language and code. However, it remains unclear how they can process grammatical syntax. In fact, under standard complexity conjectures, standard transformers cannot recognize context-free languages (CFLs), a canonical formalism to describe syntax, or even regular languages, a subclass of CFLs (Merrill et al., 2022). Merrill & Sabharwal (2025a) show that $\mathcal{O}(\log(n))$ *looping layers* (w.r.t. input length $n$) allows transformers to recognize regular languages, but the question of context-free recognition remained open. In this work, we show that *looped transformers* with $\mathcal{O}(\log(n))$ looping layers and $\mathcal{O}(n^6)$ padding tokens can recognize all CFLs. However, training and inference with $\mathcal{O}(n^6)$ padding tokens is potentially impractical. Fortunately, we show that, for natural subclasses such as unambiguous CFLs, the recognition problem on transformers becomes more tractable, requiring $\mathcal{O}(n^3)$ padding. We empirically validate our results and show that looping helps on languages that provably require logarithmic depth. Overall, our results shed light on the intricacy of CFL recognition by transformers: While general recognition may require an intractable amount of padding, natural constraints such as unambiguity yield efficient recognition algorithms.

## 1 INTRODUCTION

Transformers are proficient at many natural language (Qin et al., 2024) and coding (Jiang et al., 2024) tasks, both of which involve processing hierarchical structures. Classically, the ability to process hierarchically nested structures is closely connected to the ability to model context-free languages (CFLs). Analysis of internal representations—syntactic probing—has shown that transformers learn to encode syntactic features relevant for parsing, the task of extracting the syntactic structure of a sentence (Hewitt & Manning, 2019; Arps et al., 2022; Zhao et al., 2023). However, it is unclear what classes of syntax transformers can *provably* represent, and how CFL recognition can be implemented internally. To this end, we study whether transformers can correctly determine the grammaticality of a sentence according to a context-free grammar.

The problem of determining whether an input is grammatical can be stated as the *recognition problem for context-free grammars* (CFGs): Given a CFG $\mathcal{G}$, can a string $w$ be generated by $\mathcal{G}$? Several foundational *serial* parsing algorithms (Earley, 1970; Cocke, 1969; Kasami, 1965; Younger, 1967) solve this problem. However, such serial procedures cannot be naturally implemented by transformers due to their highly parallel, fixed-depth structure. Even regular languages, a strict subset of CFLs, cannot be recognized by fixed-depth transformers under the standard complexity conjecture $\mathrm{TC}^0 \subsetneq \mathrm{NC}^1$: Regular language recognition is complete for $\mathrm{NC}^1$ (Barrington & Thérien, 1988) while fixed-depth transformers fall in $\mathrm{TC}^0$ (Merrill et al., 2022; Chiang, 2025). *Looping* layers help: $\log(n)$ looping layers (where $n$ is the input length) allow transformers to recognize regular languages (Merrill & Sabharwal, 2025a). However, the question of whether logarithmic looping enables CFL recognition remains. In this work, we address it by analyzing the difficulty of recognizing various CFL classes by transformers. We conceptualize the difficulty in terms of extra resources needed: Looping layers and appending blank *padding* tokens (Merrill & Sabharwal, 2025b).

While general CFL recognition *cannot* be implemented by fixed-depth transformers under standard complexity conjectures, our first result shows via a direct construction that it can be expressed by looping layers $\mathcal{O}(\log(n))$ times and with $\mathcal{O}(n^6)$ padding tokens. To the best of our knowledge, this constitutes the first proof of general CFL recognition by transformers. We then ask whether simpler classes of CFLs can be recognized by transformers with fewer resources. We find that the answer is

affirmative: We show that natural subclasses of CFLs can be recognized by simpler transformers. In particular, we identify *unambiguity* and *linearity* as two key properties that make CFL recognition more tractable. Unambiguous CFLs, characterized by strings having at most one possible parse, allow for recognition with reduced padding but more looping. This aligns with transformers' struggles to parse ambiguous grammars in practice (Khalighinejad et al., 2023). Furthermore, additionally imposing linearity (where each grammar rule has at most one non-terminal on its right-hand side) reduces the amount of looping and padding required for recognizing unambiguous CFLs. We empirically test when looping helps generalization and find it to increase the performance on a log-depth complete CFL, namely the language of variable-free Boolean formulas (Buss, 1987).

In summary, we leverage theory on parallel recognition of CFLs to show that logarithmically-looped transformers can recognize CFLs, characterizing the padding requirements for different relevant subclasses. These results imply that, in order to recognize CFLs, transformers require exponentially less depth than what would be needed to implement a serial parsing algorithm like CKY. While this comes with increased space (padding) requirements in the general case, the space can be reduced for natural CFL subclasses. These results are summarized in Tab. 1.

| Language class | Padding tokens required | Looping layers required |
|---|---|---|
| General CFLs | $\mathcal{O}(n^6)$ | $\mathcal{O}(\log(n))$ |
| Unambiguous CFLs | $\mathcal{O}(n^3)$ | $\mathcal{O}(\log^2(n))$ |
| Unambiguous linear CFLs | $\mathcal{O}(n^2)$ | $\mathcal{O}(\log(n))$ |

Table 1: The computational resources required by transformers to recognize different classes of context-free languages (CFLs).

## 2 PRELIMINARIES

An **alphabet** $\Sigma$ is a finite, non-empty set of **symbols**. A **string** is a finite sequence of symbols from $\Sigma$. The **Kleene closure** $\Sigma^*$ of $\Sigma$ is the set of all strings over $\Sigma$, and $\varepsilon$ denotes the empty string. A **formal language** $\mathbb{L}$ over $\Sigma$ is a subset of $\Sigma^*$, and a **language class** is a set of formal languages.

### 2.1 CONTEXT-FREE GRAMMARS

**Definition 2.1.** *A **context-free grammar** (CFG) $\mathcal{G}$ is a tuple $(\Sigma, \mathcal{N}, S, \mathcal{P})$ where: (1) $\Sigma$ is an alphabet of **terminal** symbols (2) $\mathcal{N}$ is a finite non-empty set of **nonterminal** symbols with $\mathcal{N} \cap \Sigma = \emptyset$ (3) $\mathcal{P} \subseteq \mathcal{N} \times (\mathcal{N} \cup \Sigma)^*$ is a set of **production** rules of the form $A \to \boldsymbol{\alpha}$ for $A \in \mathcal{N}$ and $\boldsymbol{\alpha} \in (\Sigma \cup \mathcal{N})^*$ (4) $S \in \mathcal{N}$ is a designated start non-terminal symbol. As standard, we denote terminal and nonterminal symbols by lowercase and uppercase symbols, respectively.*

A sequence of non-terminals and terminals $\boldsymbol{\alpha} \in (\mathcal{N} \cup \Sigma)^*$ is a **sentential form**. A CFG generates strings by repeatedly applying rules to sentential forms derived from the start symbol until it produces a sequence of terminal symbols, i.e., a **string**. We call this procedure a **derivation**, and the resulting string its **yield**. We define the relation $A \to \beta$ if $\exists p \in \mathcal{P}$ such that $p = (A \to \boldsymbol{\alpha}\boldsymbol{\beta}\boldsymbol{\gamma})$ where $\boldsymbol{\alpha}, \boldsymbol{\beta}, \boldsymbol{\gamma}$ are sentential forms. We denote by $\xrightarrow{*}$ the reflexive, transitive closure of $\to$.

**Definition 2.2.** *The **language of a CFG** $\mathcal{G}$ is the set $\mathbb{L}(\mathcal{G}) \overset{\text{def}}{=} \{\boldsymbol{w} \in \Sigma^* \mid S \xrightarrow{*} \boldsymbol{w}\}$.*

**Definition 2.3.** *A language $\mathbb{L}$ is **context-free** if there exists a CFG $\mathcal{G}$ such that $\mathbb{L}(\mathcal{G}) = \mathbb{L}$.*

It is common practice to consider CFGs in a normal form, namely:

**Definition 2.4.** *A CFG $\mathcal{G}$ is in **Chomsky Normal Form (CNF)** if any $p \in \mathcal{P}$ is either of the form $A \to BC$, $A \to a$ or $S \to \varepsilon$.*

Every CFG can be transformed into an equivalent one in CNF.

## 2.2 TRANSFORMERS

We consider the idealization of transformers from Merrill & Sabharwal (2025a;b). In short,[1] we study **average hard attention** transformers (AHATs), where the attention normalization function returns a uniform average of the values of tokens that maximize the attention score. The transformers use *multi*-pre-norm, where the layer normalization is applied before the residual connection on either the entire hidden state or on distinct subsets thereof (Merrill & Sabharwal, 2024). We further assume logarithmic-precision arithmetic, where computations are performed with $\mathcal{O}(\log(n))$ bits for an input of size $n$. Coupling AHATs and log-precision unlocks useful gadgets such as storing string indices, counting symbol occurrences across the string and performing equality checks of values stored in residual streams at separate positions (Merrill & Sabharwal, 2024). We assume input strings to the transformer are augmented with both a beginning-of-sequence (BOS) and end-of-sequence (EOS) token. Denote by $\boldsymbol{x}_{\text{EOS}}^{L}$ the contextual representation of EOS at end of the forward pass of the transformer. We apply a linear classifier to $\boldsymbol{x}_{\text{EOS}}^{L}$ to determine string acceptance.

Looped transformers scale the number of layers with input length (Merrill & Sabharwal, 2025a).

**Definition 2.5.** *Let T be a transformer. We denote by $\langle A, B, C \rangle$ a partition of layers such that $A$ is the **initial block** of layers, $B$ is the **looped** block of layers and $C$ is the **final block** of layers. T is $d(n)$-**looped** if upon a forward pass with an input of length $n$, $B$ is repeated $\mathcal{O}(d(n))$ times.*

The amount of computation performed by self attention is definitionally quadratic in the string length. One can dynamically increase this by adding *padding space* (Merrill & Sabharwal, 2025b).

**Definition 2.6.** *Let T be a transformer. T is $w(n)$-**padded** if $\mathcal{O}(w(n))$ padding tokens are appended to the end of the string when computing the contextual representations of a length-$n$ input.*

Scaling number of layers and padding tokens in transformers is analogous to scaling time and space Boolean circuits (Merrill & Sabharwal, 2025b), a classical parallel model of computation. Allowing for different looping and padding budgets results in different classes of transformers. We adopt naming conventions of these models from Merrill & Sabharwal (2025b). We denote by $\mathsf{AHAT}_k^d$ the class of languages recognized by averaging hard-attention transformers with $\mathcal{O}(\log^d(n))$ looping, $\mathcal{O}(n^k)$ padding and strict causal masking. We further denote with uAHAT average hard-attention transformers with no masking, and with mAHAT transformers that use both masked and unmasked attention heads. Conveniently, AHATs can simulate uAHATs:

**Lemma 2.1** (Merrill & Sabharwal 2025b Proposition 1.). $\mathsf{uAHAT}_k^d \subseteq \mathsf{mAHAT}_k^d \subseteq \mathsf{AHAT}_{1+\max(k,1)}^d$ *for $d \geq 1$.*

# 3 RECOGNIZING GENERAL CFLS WITH TRANSFORMERS

We now describe a parallel algorithm for general CFL recognition, which synthesizes ideas from previous work on algorithms for parallel CFL recognition (Ruzzo, 1980; Rossmanith & Rytter, 1992; Lange & Rossmanith, 1990). We then show how to implement this algorithm on AHATs, allowing us to prove the following theorem:

**Theorem 3.1.** *Given a CFL $\mathbb{L}$, there exists a transformer with both causally-masked and non-masked attention layers, $\mathcal{O}(\log(n))$ looping layers and $\mathcal{O}(n^6)$ padding tokens that recognizes $\mathbb{L}$. That is, $\mathsf{CFL} \subseteq \mathsf{mAHAT}_6^1 \subseteq \mathsf{AHAT}_7^1$.*

Our goal is to recognize a CFL represented by a grammar in CNF (Def. 2.4) with start symbol S. For a string $\boldsymbol{w}$ of length $n$, the algorithm determines whether $\boldsymbol{w} \in \mathbb{L}(\mathcal{G})$. To do this, it manipulates **items**—tuples of the form $[A, i, j]$, where $A \in \mathcal{N}$ and $i, j \in [n] \stackrel{\text{def}}{=} \{1, 2, \ldots, n\}$. The item $[A, i, j]$ is **realizable** if and only if $A \stackrel{*}{\to} w_i w_{i+1} \ldots w_j$, i.e., if there is a sequence of rules that can be applied to the non-terminal A that yields $w_i w_{i+1} \ldots w_j$.

We further define **slashed** items of the form $[A, i, j]/[B, k, l]$, where $i \leq k \leq l \leq j$. Intuitively, solving $[A, i, j]/[B, k, l]$ equates to determining whether A can derive $w_i \ldots B \ldots w_j$ *assuming* that the non-terminal B already derives the substring $w_k \ldots w_l$. More formally, $[A, i, j]/[B, k, l]$ is **realizable** if and only if $A \stackrel{*}{\to} w_i w_{i+1} \ldots w_{k-1} B w_{l+1} \ldots w_j$.

---

[1] We refer to App. A for more details on the transformer model.

Naturally, $\boldsymbol{w} \in \mathbb{L}(\mathcal{G})$ if and only if the item $[\mathrm{S}, 1, n]$ is realizable, and determining realizability can be broken down recursively as follows:

**Lemma 3.1.** $[\mathrm{X}, i, j]$ *is realizable if and only if one of the following conditions is met:*

- ***Base case:*** $j = i$ *and* $\mathrm{X} \to \mathtt{w}_i$ *is a rule in the grammar for some* $\mathtt{w}_i$.
- ***Recursive case 1:*** *There exist a rule* $\mathrm{X} \to \mathrm{YZ}$ *and an index* $k$ *such that* $[\mathrm{Y}, i, k-1]$ *and* $[\mathrm{Z}, k, j]$ *are realizable items. There are* $\mathcal{O}(|\mathcal{P}|n)$ *ways to choose a rule and an index for* $\mathcal{O}(|\mathcal{N}|n^2)$ *possible input items* $[\mathrm{X}, i, j]$.
- ***Recursive case 2:*** *There exists a* $[\mathrm{Y}, k, l]$ *such that* $[\mathrm{X}, i, j]/[\mathrm{Y}, k, l]$ *and* $[\mathrm{Y}, k, l]$ *are both realizable. There are* $\mathcal{O}(|\mathcal{N}|n^2)$ *possible items of the form* $[\mathrm{Y}, k, l]$ *for* $\mathcal{O}(|\mathcal{N}|n^2)$ *possible input items* $[\mathrm{X}, i, j]$.

*Proof.* The proof follows from our definitions. In the base case, if $j = i$, then X needs to derive exactly the symbol $\mathtt{w}_i$ in one step without producing non-terminals (assuming a CFG with no useless non-terminals). In the recursive case, if $[\mathrm{X}, i, j]$ is realizable then there exists some associated parse tree where $\mathrm{X} \xrightarrow{*} \mathtt{w}_i \ldots \mathtt{w}_j$. Such a tree can be split by selecting a split vertex which induces recursive subproblems. If the chosen split vertex is the root X, there exists a rule $\mathrm{X} \to \mathrm{YZ}$ such that Y and Z derive disjoint, consecutive substrings of $\boldsymbol{w}$. If the chosen split vertex is a non-root $\mathrm{Y} \in \mathcal{N}$, then Y derives some substring $\mathtt{w}_k \ldots \mathtt{w}_l$, and X derives $\boldsymbol{w}$ where $\mathtt{w}_k \ldots \mathtt{w}_l$ has been replaced by Y. ∎

**Lemma 3.2.** $[\mathrm{X}, i, j]/[\mathrm{Y}, k, l]$ *is realizable if and only if one of the following conditions is met:*

- ***Base case:*** $k = i$, $l = j - 1$ *and there is a rule* $\mathrm{X} \to \mathrm{YZ}$ *in the grammar such that* $\mathrm{Z} \to \mathtt{w}_j$. *(and symmetric case)*
- ***Recursive case 1:*** *There exist a rule* $\mathrm{X} \to \mathrm{AB}$ *and an index* $p$ *such that* $[\mathrm{A}, i, p-1]/[\mathrm{Y}, k, l]$ *and* $[\mathrm{B}, p, j]$ *are realizable items (and symmetric case). There are* $\mathcal{O}(|\mathcal{P}|n)$ *ways to choose a rule and an index for* $\mathcal{O}(|\mathcal{N}|^2 n^4)$ *possible input slashed items* $[\mathrm{X}, i, j]/[\mathrm{Y}, k, l]$.
- ***Recursive case 2:*** *There exists a* $[\mathrm{Z}, p, q]$ *such that* $[\mathrm{X}, i, j]/[\mathrm{Z}, p, q]$ *and* $[\mathrm{Z}, p, q]/[\mathrm{Y}, k, l]$ *are both realizable. There are* $\mathcal{O}(|\mathcal{N}|n^2)$ *possible items of the form* $[\mathrm{z}, p, q]$ *for* $\mathcal{O}(|\mathcal{N}|^2 n^4)$ *possible input slashed items* $[\mathrm{X}, i, j]/[\mathrm{Y}, k, l]$.

*Proof.* The proof follows the same structure as the proof of Lem. 3.1. In the base case, X needs to derive in one step the non-terminal Y and some non-terminal Z such that Z derives in one step a symbol at the boundary of $\mathtt{w}_i \ldots \mathtt{w}_j$ (either $\mathtt{w}_i$ or $\mathtt{w}_j$). In the recursive case, if $[\mathrm{X}, i, j]/[\mathrm{Y}, k, l]$ is realizable then there exists a parse tree associated with it where $\mathrm{X} \xrightarrow{*} \mathtt{w}_i \ldots \mathtt{w}_{k-1} \mathrm{Y} \mathtt{w}_{l+1} \ldots \mathtt{w}_j$. Such a tree can be split by selecting a split vertex which induces recursive subproblems. If the chosen split vertex is the root X, there exists a rule $\mathrm{X} \to \mathrm{AB}$ such that A derives some sentential form $\mathtt{w}_i \ldots \mathtt{w}_{k-1} \mathrm{Y} \mathtt{w}_{l+1} \ldots p$ and B derives the string $\mathtt{w}_{p+1} \ldots \mathtt{w}_j$ for some index $p \in [n]$. If the chosen split vertex is a non-root $\mathrm{Z} \in \mathcal{N}$, then Z derives the sentential form $\mathtt{w}_p \ldots \mathtt{w}_{k-1} \mathrm{Y} \mathtt{w}_{l+1} \ldots \mathtt{w}_q$ and X derives the sentential form $\mathtt{w}_i \ldots \mathtt{w}_{p-1} \mathrm{Z} \mathtt{w}_{q+1} \ldots \mathtt{w}_j$ for some indices $p, q \in [n]$. ∎

**Parallel algorithms for CFL recognition.** Lemmata 3.1 and 3.2 state that an item is realizable if it can be decomposed into realizable subproblems. Rather than enumerating all the possible decompositions sequentially, we will leverage parallelism to simultaneously compute the realizability of all the induced subproblems. The term *guessing* has been coined (Ruzzo, 1980) to denote the ability of a parallel model of computation to attend to a valid computation path given an unbounded set of possible computations. By analogy, we can *guess* which of the correct decompositions of an item is correct by leveraging parallelism, and then recursively verify the induced subproblems in parallel. This suggests natural parallel algorithms for checking realizability, which we present in Algs. 1 and 2.

---

**Algorithm 1** Determining if the item $[X, i, j]$ is realizable.

1. **def** SOLVE($[X, i, j]$):
2.   **if** $i = j$ :
3.     **return** $X \to \mathsf{w}_i \in \mathcal{P}$
4.   **guess** an integer $x \in \{1, 2\}$
5.   **if** $x = 1$ :
6.     **guess** a rule $X \to YZ \in \mathcal{P}$ and $k \in [n]$
7.     **return** SOLVE($[Y, i, k-1]$) $\wedge$ SOLVE($[Z, k, j]$)
8.   **else**
9.     **guess** an item $[Y, k, l]$
10.     **return** SOLVE($[X, i, j]/[Y, k, l]$) $\wedge$ SOLVE($[Y, k, l]$)

---

**Algorithm 2** Determining if the item $[X, i, j]/[Y, k, l]$ is realizable.

1. **def** SOLVE($[X, i, j]/[Y, k, l]$):
2.   **if** $k = i \wedge l = j - 1$ :
3.     **return** $\exists\, Y, Z \in \mathcal{N}$ such that $X \to YZ \in \mathcal{P} \wedge Z \to \mathsf{w}_j \in \mathcal{P}$
4.   **guess** an integer $x \in \{1, 2\}$
5.   **if** $x = 1$ :
6.     **guess** a rule $X \to AB \in \mathcal{P}$ and $p \in [n]$
7.     **return** SOLVE($[A, i, p-1]/[Y, k, l]$) $\wedge$ SOLVE($[B, p, j]$)
8.   **else**
9.     **guess** an item $[Z, p, q]$
10.     **return** SOLVE($[X, i, j]/[Z, p, q]$) $\wedge$ SOLVE($[Z, p, q]/[Y, k, l]$)

---

Intuitively, the recursive function SOLVE defined in Algs. 1 and 2 computes the realizability of items.

**Theorem 3.2** (Correctness). *Given a CFG $\mathcal{G}$ in CNF and $\boldsymbol{w} \in \Sigma^*$ of length $n$, SOLVE($[S, 1, n]$) = 1 if and only if $\boldsymbol{w} \in \mathbb{L}(\mathcal{G})$.*

*Proof.* By definition, $\boldsymbol{w} \in \mathbb{L}(\mathcal{G})$ if and only if $[S, 1, n]$ is realizable. By Lemmata 3.1 and 3.2, the item $[S, 1, n]$ is realizable if and only if there exists a decomposition of $[S, 1, n]$ that respects Lemmata 3.1 and 3.2. SOLVE recursively guesses such decompositions, guaranteeing that we will compute a valid decomposition if it exists. ∎

We now analyze the resources required to compute SOLVE$[S, 1, n]$, which is equivalent to testing membership of the input string $\boldsymbol{w}$ in the given grammar $\mathcal{G}$. The recursive procedure induced by SOLVE is based on a balanced decomposition of problems into subproblems of roughly equal size, which intuitively leads to a $\log(n)$-time procedure. Formally, we have the following well-known theorem for decomposing trees:

**Theorem 3.3** (Jordan 1869). *Given a tree with $n$ vertices, there exists a vertex whose removal partitions the tree into two trees with each at most $n/2$ vertices.*

We rely on Thm. 3.3 to prove that Alg. 1 runs in a logarithmic number of recursive steps:

**Theorem 3.4.** *We can compute SOLVE($[S, 1, n]$) in $\log(n) + \mathcal{O}(1)$ recursive steps $\forall \boldsymbol{w} \in \Sigma^*$ with $|\boldsymbol{w}| = n$.*

*Proof.* By Thm. 3.3, for any realizable item, there exists a balanced decomposition of the corresponding parse tree into two trees of roughly equal size which can be represented by two items (the split is at the root) or a slashed item and an item (the split is not at the root). Assuming we can process all possible tree decompositions in parallel, we will necessarily guess the balanced one where subtrees have at most $2n/2 + 1$ vertices (a full binary tree with $n$ leaves does not have more than $2n$ vertices). After $i$ recursive steps, the current subtrees have at most $\frac{n}{2^{i-1}} + \mathcal{O}(1)$ vertices. Therefore, we will solve all base cases after at most $\log(n) + \mathcal{O}(1)$ steps. ∎

**Space complexity.**    The bottleneck resides in solving an item $[X, i, j]/[Y, k, l]$, which occupies $\mathcal{O}(n^4)$ space, and guessing an item $[Z, p, q]$ that could decompose this problem, which itself occupies $\mathcal{O}(n^2)$ space, leading to a total space complexity of $\mathcal{O}(n^6)$.

Combining both insights on time- and space-complexity, we can prove the following theorem:

**Theorem 3.1.** *Given a* CFL $\mathbb{L}$*, there exists a transformer with both causally-masked and non-masked attention layers,* $\mathcal{O}(\log(n))$ *looping layers and* $\mathcal{O}(n^6)$ *padding tokens that recognizes* $\mathbb{L}$*. That is,* CFL $\subseteq$ mAHAT$_6^1$ $\subseteq$ AHAT$_7^1$.

*Proof intuition.*    The construction implements Algs. 1 and 2 on a transformer. Intuitively, each item and possible decomposition is associated with a padding token. There are $\mathcal{O}(n^6)$ ways to enumerate items and a possible decomposition. We assume a three-value logic system, where each item is associated with a value in $\{0, 1, \perp\}$ to denote that the item is unrealizable (0), realizable (1) or not known yet to be realizable ($\perp^2$). Each padding token allocates space for this value. Intuitively, we will develop a construction such that padding tokens compute the information of whether their associated item is realizable w.r.t. the given decomposition. Initially, all padding tokens store $\perp$. In the initial block of layers, padding tokens associated with a base case item of the form $[A, i, i]$ can attend to symbol representations via an equality-check to verify whether the base case is valid, i.e., $A \rightarrow w_i \in \mathcal{P}$. In the inductive step, padding tokens attend to the padding tokens associated with the decomposition via an equality-check. A feedforward network then either adds 1 to the residual stream if both sub-items are realizable, 0 if any of them is non-realizable, or $\perp$ if realizability can not be determined at the current iteration. It takes $\log(n)$ looping layers to populate the values of all items in their respective padding tokens due to Thm. 3.3. Finally, we can check whether there exists a padding token associated with $[S, 1, n]$ that holds the value 1. Applying Lem. 2.1 yields inclusion in AHAT$_7^1$. The detailed proof is in App. B.1. ∎

## 4    UNAMBIGUITY REDUCES PADDING REQUIREMENTS FOR RECOGNITION

§3 shows that $\log(n)$-depth mAHATs with $\mathcal{O}(n^6)$ padding can recognize all CFLs. The large amount of padding is undesirable, but somewhat necessary—intuitively, an algorithm for recognizing an arbitrary CFL requires a large amount of padding because the grammar can be highly ambiguous. Guessing how to decompose an arbitrary item requires a substantial amount of space. Accordingly, we next study *unambiguous* CFLs and show that they require less padding by proving the following theorem.

**Theorem 4.1.** *Let* UCFL *be the classes of unambiguous* CFLs*. Then* UCFL $\subseteq$ mAHAT$_3^2$ $\subseteq$ AHAT$_4^2$.

A CFL is **unambiguous** if there is at most one possible derivation (i.e, parse tree) for any string. Unambiguity is a natural CFL feature of general interest. Transformers struggle to parse ambiguous grammars (Khalighinejad et al., 2023) and struggle to process syntactically ambiguous natural language sentences (Liu et al., 2023). Moreover, modern parsers for programming languages such as LR parsers rely on deterministic (therefore unambiguous) CFLs to process inputs in linear time.

This section first introduces an unambiguous CFG recognition algorithm with a tractable space complexity in $\log^2(n)$-time. We then translate this algorithm into AHATs with a tractable amount of padding.

### 4.1    A PATH SYSTEM FRAMEWORK FOR UNAMBIGUOUS CFL RECOGNITION

We formulate recognition of unambiguous CFLs as a **path system** problem. A path system consists of initial vertices that are associated with either the value 1 or 0, and a relation $\mathcal{R}$ that formalizes how to connect the vertices. By associating base case items of the form $[A, i, i]$ to initial vertices, general items of the form $[A, i, j]$ to arbitrary vertices, and connecting vertices depending on the rules of the given grammar, we can compute the realizability of an item by finding a path between its associated vertices and a base node. We now present Chytil et al. (1991)'s path system framework for recognizing unambiguous CNF CFGs and express it in AHATs.

---

[2]We write $\perp$ for ease of notation. Concretely, $\perp$ can be encoded as any integer that is neither 0 nor 1.

We denote by $\mathcal{V}$ a set of vertices, each associated with a tuple $[A, i, j]$. We denote by $\mathcal{T} \subseteq \mathcal{V}$ the **initial** set of vertices of the form $[A, i, i]$ such that $A \to w_i \in \mathcal{P}$. $\mathcal{R}(x, y, z) : \mathcal{V}^3 \to \{0, 1\}$ is a function that relates how to connect the vertices, where $\mathcal{R}(x, y, z) = 1$ if and only if $z$ is associated with some tuple $[A, i, j]$, $x$ is associated with some tuple $[B, i, k]$, and $y$ is associated with some tuple $[C, k, j]$ such that $A \to BC \in \mathcal{P}$. We denote by $\mathcal{C}(w) \subseteq \mathcal{V}$ the smallest set containing $\mathcal{T}$ such that if $x, y \in \mathcal{C}(w)$ and $\mathcal{R}(x, y, z) = 1$ then $z \in \mathcal{C}(w)$, i.e., $\mathcal{C}(w)$ is the closure of $\mathcal{T}$ with respect to $\mathcal{R}$. One can intuitively think of $\mathcal{C}(w)$ as the set of realizable elements, and the recognition problem is thus equivalent to determining whether the vertex associated with $[S, 1, n]$ is in the set $\mathcal{C}(w)$.

Let us now describe how to compute $\mathcal{C}(w)$. Let $\mathcal{X} \subseteq \mathcal{V}$ be a set of **marked** vertices. A **dependency graph** with respect to $\mathcal{X}$, denoted $DG(\mathcal{X})$, is the directed graph $\mathcal{G} = (\mathcal{V}, \mathcal{E})$ where:

$$\mathcal{E} = \{(z, x) \mid z \notin \mathcal{X}, \mathcal{R}(x, y, z) = 1 \text{ or } \mathcal{R}(y, x, z) = 1 \text{ for some } y \in \mathcal{X}\} \tag{1}$$

Intuitively, assuming $\mathcal{X} \subseteq \mathcal{C}(w)$, the edge $(z, x)$ can be interpreted as follows: $x \in \mathcal{C}(w)$ implies that $z \in \mathcal{C}(w)$. Precisely, $(z, x)$ being an edge signals that there is some vertex $y$ associated with a realizable item such that $\mathcal{R}(x, y, z) = 1$ or $\mathcal{R}(y, x, z) = 1$. Therefore, if $x$ is also associated with a realizable item (i.e, is in the closure $\mathcal{C}(w)$), then $z$ is a realizable item. The algorithm iteratively expands the known set of vertices to be associated with realizable items by computing the set of vertices that have a directed path to a marked node. We denote by REACH($\mathcal{D}$) the vertices of the dependency graph $\mathcal{G}$ that have a directed path to a marked vertex in $\mathcal{D}$. Chytil et al. (1991)'s procedure to compute $\mathcal{C}(w)$ is described in Alg. 3.

---

**Algorithm 3** Algorithm for computing $\mathcal{C}(w)$

1. **def** COMPUTE CLOSURE($w, \mathcal{G}$):
2.    **initialize** $\mathcal{V} \leftarrow \{[A, i, j]\}$
3.    **initialize** $\mathcal{T} \leftarrow \{[A, i, i] \mid A \to w_i \in \mathcal{P}\}$
4.    $\mathcal{X} \leftarrow \mathcal{T}$
5.    **for** _ in range $\log(n)$ :
6.       $\mathcal{D} \leftarrow DG(\mathcal{X})$
7.       $\mathcal{X} \leftarrow$ REACH($\mathcal{D}$)
8.    **return** $\mathcal{X}$

---

The bottleneck in Alg. 3 is computing REACH($\mathcal{D}$), i.e., reachability queries on a directed, acyclic graph (DAG). Assuming unambiguity, we have the following powerful insight: There is at most one path between any pair of vertices in the DAG. By contradiction, if there are multiple paths from a vertex $[A, i, j]$ to another vertex $[B, k, l]$ there are then different derivations that can reduce $[A, i, j]$ to $[B, k, l]$, which contradicts the unambiguity condition. Therefore, for each vertex $v$, the subgraph induced by vertices reachable from $v$ becomes a *tree* rooted at $v$. Reachability queries on a tree reduce to evaluating the corresponding *Boolean formula*, where leaf vertices are assigned 1 if they correspond to realizable items and non-leaf vertices are assigned the $\vee$ operator. We rely on the following lemma to perform this procedure:

**Lemma 4.1.** *Let $\psi$ be a variable-free Boolean formula. Assume $\psi$ is represented in a transformer's residual stream as follows, where we consider the binary tree induced by $\psi$. For each leaf, there is a padding token that encodes its value (1 or 0). For each function node, there is a padding token that encodes its type ($\wedge$ or $\vee$) and pointers to its input arguments. Then, we can compute the value of each subformula in $\mathcal{O}(\log(n))$ time on an input of length $n$.*

*Proof intuition.* Given the appropriate pointers, we implement Rytter (1985)'s parallel pebble game algorithm for evaluating Boolean formulas with $\mathcal{O}(\log(n))$ steps on transformers. Each vertex $v$ allocates space in its residual stream for 1) a VALUE corresponding to the evaluation of $v$'s associated formula 2) a pointer to some descendant vertex PTR of $v$ 3) a conditional function CONDF: $\{0, 1\} \to \{0, 1\}$ based on the current vertex type ($\wedge$ or $\vee$). The intuition of PTR is that if we know PTR.VALUE, we can evaluate the current node's value via the conditional function CONDF(PTR.VALUE). The procedure operates in parallel at each vertex by iterating three steps $\mathcal{O}(\log(n))$ times: activate, square, and pebble. Rytter (1985) shows that this algorithm correctly evaluates each subformula in $\mathcal{O}(\log(n))$ steps. The detailed proof is in App. B.2.

$\blacksquare$

We can now show how to simulate Alg. 3's procedure on transformers for unambiguous CFLs with $\mathcal{O}(\log(n)^2)$ looping layers and $\mathcal{O}(n^3)$ padding tokens.

**Theorem 4.1.** *Let UCFL be the classes of unambiguous CFLs. Then UCFL $\subseteq$ mAHAT$_3^2$ $\subseteq$ AHAT$_4^2$.*

*Proof intuition.* We implement Alg. 3 on mAHATs. Each item $[A, i, j]$ (of which there are $\mathcal{O}(n^2)$) is assigned a padding token. For each item $[A, i, j]$, there are $\mathcal{O}(n)$ ways to decompose it using a split index $k \in [n]$. For every potential edge between vertices associated with $[A, i, j]$ and some $[B, i, k]$ (or $[B, k, j]$), we assign a padding token. As in Thm. 3.1, we assume a three-valued logic system where padding tokens for vertices are at any step assigned an element in $\{0, 1, \perp\}$, denoting non-realizability (0), realizability (1) or not yet known to be realizable ($\perp$). Initially, all padding tokens store $\perp$.

Initially, padding tokens for vertices can check whether they are associated with base case items of the form $[A, i, i]$. These padding tokens can add to their residual stream 1 (item is realizable) or 0 (item is non-realizable) depending on if $A \to w_i \in \mathcal{P}$.

In the iterative case, each padding token for an edge associated with items $[A, i, j], [B, i, k]$ can first check whether there exists a rule $A \to BC$ and if so, add to the residual stream $[C, k+1, j]$. Crucially, there are finitely such items (proportional to $|\mathcal{N}|$ as the splitting index $k$ is fixed). Padding token for edges can attend to padding tokens associated with $[C, k+1, j]$ and check whether any of them stores 1, denoting realizability. In that case, the padding token associated with $[A, i, j], [B, i, k]$ signals that the edge $([A, i, j], [B, i, k])$ is now in the graph (following how we define edges in Eq. (1)). Padding tokens for vertices associated with items $[A, i, j]$ can therefore attend to padding tokens for edges associated with $[A, i, j], [B, i, k]$, which yields the dependency graph.

Crucially, due to unambiguity, for each vertex $v$, the subgraph induced by vertices reachable from $v$ becomes a tree rooted at $v$. We then show how to binarize this tree. Reachability queries on a binary tree can be reduced to the evaluation of the induced Boolean formula (Chytil et al., 1991). We invoke Lem. 4.1 to evaluate Boolean formulas in $\log(n)$ steps. The detailed proof is in App. B.2. ∎

## 4.2 Unambiguous linear CFLs require less time and space

Finally, we show how **linearity** further reduces the resources needed by transformers to recognize unambiguous CFLs. A **linear** CFL is one recognized by a CFG where each rule is the form $A \to aB$, $A \to Ba$, or $A \to a$. While restricted, linear CFLs capture a wide range of features of context-freeness. For example, *balanced counting* can be modeled by the linear CFL $\mathbb{L} = \{a^n b^n \mid n \geq 0\}$, and *symmetry* can be modeled by the linear CFL $\mathbb{L} = \{ww^R \mid w \in \Sigma^*\}$.

We consider unambiguous linear[3] CFLs (ULCFLs) and show they can be recognized by $\log$-depth transformers with quadratic padding.

**Theorem 4.2.** $\text{ULCFL} \subseteq \text{mAHAT}_2^1 \subseteq \text{AHAT}_3^1$.

*Proof.* We implement Alg. 3 on AHATs and show how linearity reduces the computational requirements w.r.t. Thm. 4.1. We define $\mathcal{V}$ and $\mathcal{T}$ as in §4.1. Assuming linearity, there is an edge from $v_1$ to $v_2$ if and only if $v_1$ takes the form $[A, i, j]$, $v_2$ takes the form $[B, i+1, j]$ such that $A \to w_i B \in \mathcal{P}$ (or the symmetric case). We first remark that we now have a *constant* number of outgoing edges for each node. Due to linearity, rules that spawn non-terminals are of the form $A \to wB$ or $A \to Bw$, and solving an item $[A, i, j]$ therefore reduces to solving items that aim to derive either $w_{i+1} \ldots w_j$ or $w_i \ldots w_{j-1}$. There are finitely many such items given $[A, i, j]$ as the indices are fixed. Therefore, the procedure can be implemented with $\mathcal{O}(n^2)$ padding tokens.

Moreover, because every production rule now necessarily spawns a terminal symbol, the full dependency graph can be constructed via $\text{DG}(\mathcal{T})$. If $A \to wB$ is a production rule used in the derivation of a string, then $[w, i, i] \in \mathcal{T}$ for some $i$, and $\mathcal{R}([w, i, i], [B, i+1, j], [A, i, j]) = 1$. Crucially, any production rule applied in the derivation of a string that reduces some item $[A, i, j]$ to another item $[B, i+1, j]$ leads to an edge between their associated items in the *initial* dependency graph $\text{DG}(\mathcal{T})$. Therefore, we can compute the realizability of all items with a single call to REACH on the initial dependency graph $\text{DG}(\mathcal{T})$, and $\log(n)$ looping layers then suffice to perform Alg. 3. ∎

---

[3]There is a subtlety here: A CFL can be induced by both a non-linear unambiguous grammar and by a different linear, ambiguous grammar. Here we consider grammars that are *simultaneously* linear and unambiguous.

## 5 EXPERIMENTS

We conduct experiments to elicit the impact of looping when recognizing formal languages, and provide more details on our experimental setup in App. C. We train transformer classifiers on CFLs of varying degrees of complexity:

- **Boolean formula value problem** (BFVP): The set of variable-free Boolean formulas that evaluate to 1. This CFL is known to be complete for $NC^1$ (Buss, 1987), i.e., requires logarithmic time w.r.t. input length. We consider formulas in the standard *infix* notation (e.g., $1 \vee 0$ is in infix notation) as well as *postfix* notation (e.g., $1\ 0 \vee$ is in postfix notation). Parallel algorithms for BFVP typically rely on postfix notation (Buss, 1987; Buss et al., 1992).
- **Palindrome**: The language $\mathbb{L} = \{ww^R \mid w \in \Sigma^*\}$ for some alphabet $\Sigma$. We focus on a binary alphabet. This language is linear unambiguous and non-deterministic. Prior work has shown that fixed-depth transformers with hard attention can recognize this language (Hao et al., 2022).
- **Marked Palindrome**: This language simplifies Palindrome by extending strings with a marker between $w$ and $w^R$, which delimits at which index we reverse the string. In other words, $\mathbb{L} = \{w\#w^R \mid w \in \Sigma^*\}$ where $\# \notin \Sigma$. This language is linear deterministic.
- **Dyck**: The language of nested strings of parentheses of $k$ types, which we denote by $D(k)$. We consider $D(1)$ and $D(2)$. This language is non-linear and deterministic. Fixed-depth transformers can recognize $D(k)$ for any $k$ (Weiss et al., 2021).

These languages vary in complexity, allowing us to test transformers' ability to learn CFL recognition constructions for languages of different difficulties. In particular, while Palindrome and $D(k)$ languages can in principle be recognized by constant-depth transformers, BFVP requires growing depth (i.e., log-depth), assuming $TC^0 \neq NC^1$. This suggests that the performance of log-depth vs. constant-depth transformers on BFVP is a good measure of whether transformers can utilize the extra expressivity of log-depth when it is required. Our results are presented in Tab. 2.

Table 2: Mean accuracy ($\pm$ standard deviation) by language and transformer type across seeds.

| Language | Test accuracy on in-distribution strings | | Test accuracy on out-of-distribution strings | |
|---|---|---|---|---|
| | Fixed-depth | $\log(n)$ looping | Fixed-depth | $\log(n)$ looping |
| BFVP | $0.97 \pm 0.01$ | $0.98 \pm 0.00$ | $0.88 \pm 0.01$ | $0.91 \pm 0.01$ |
| BFVP (postfix) | $0.95 \pm 0.01$ | $0.98 \pm 0.00$ | $0.87 \pm 0.01$ | $0.91 \pm 0.01$ |
| Palindrome | $0.94 \pm 0.01$ | $0.93 \pm 0.01$ | $0.79 \pm 0.03$ | $0.72 \pm 0.03$ |
| Marked palindrome | $0.97 \pm 0.01$ | $0.98 \pm 0.01$ | $0.59 \pm 0.19$ | $0.66 \pm 0.18$ |
| $D(1)$ | $0.98 \pm 0.00$ | $0.98 \pm 0.00$ | $0.94 \pm 0.02$ | $0.93 \pm 0.01$ |
| $D(2)$ | $0.98 \pm 0.02$ | $0.99 \pm 0.00$ | $0.83 \pm 0.08$ | $0.90 \pm 0.08$ |

**Results.** Despite our theoretical analysis, the difference in performance between looped- and non-looped transformers is not stark, which can be explained by the fact that most of the languages we test transformers on have fixed-depth solutions. We conjecture looping should not offer a substantial gain in performance for problems where fixed-depth solutions suffice. Moreover, we remark that for both variants of BFVP, looping leads to slight improvements in in-distribution (1-3%) and generalization (3-4%) accuracy. This result is consistent with the fact that BFVP is known to require log-depth. For Palindrome and $D(1)$, looping does not improve accuracy, which is supported by the fact that these languages already have fixed-size solutions (Hao et al., 2022; Weiss et al., 2021). For $D(2)$ and Marked Palindrome, looping seems to improve generalization even though these languages also have constant-depth transformer constructions.

## 6 DISCUSSION AND CONCLUSION

We show that transformers with $\log$-depth can recognize general CFLs if they can use padding tokens (Merrill & Sabharwal, 2025b). In addition, we characterize unambiguity and linearity as CFL features that can reduce the amount of padding needed by transformers for recognition. These results reveal one way that transformers with limited depth can recognize CFLs and predict ambiguity in language could be a hurdle for transformers to process, as suggested in previous empirical work

(Khalighinejad et al., 2023; Liu et al., 2023). Although it is not possible to improve our $\log$-depth recognition algorithm to fixed depth unless $\text{TC}^0 = \text{NC}^1$, our padding bounds are not known to be tight. Therefore, future work could find more padding-efficient transformer constructions for recognizing general CFLs, or subclasses thereof. Additionally, it would be interesting to consider the psycholinguistic implications of our results for comparing how humans and LMs process language and syntax. It is believed that CFLs are too weak to model natural language (Shieber, 1988), and that mildly context-sensitive formalisms such as tree-adjoining grammars (TAGs) are a better prospect to model natural language (Joshi, 1985; Bordihn, 2004). Future work could therefore focus on analyzing transformers' ability to recognize languages induced by TAGs (TALs). Finally, because expressivity results cannot fully predict the empirical abilities of transformers, recent learnability results (Hahn & Rofin, 2024) are painting a more complete picture of the abilities and limitations of transformers. We therefore encourage future work to investigate theoretically the conditions under which a transformer can learn to process syntax on out-of-distribution inputs.

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

## A  EXTENDED BACKGROUND

### A.1  TRANSFORMER MODELS

We introduce in this section our idealization of the transformer architecture.

#### A.1.1  FIXED-SIZE TRANSFORMERS

An $L$-layer **transformer** of constant width [4] $D$ is a mapping $\mathsf{T}\colon \Sigma^* \to \left(\mathbb{R}^D\right)^*$:

$$\mathsf{T} \stackrel{\text{def}}{=} \mathcal{L}^{(L)} \circ \cdots \circ \mathcal{L}^{(1)} \circ \mathsf{embed} \tag{2}$$

The **input encoding function** $\mathsf{embed}\colon \Sigma^* \to \left(\mathbb{R}^D\right)^*$ applies an injective position-wise embedding function to each symbol in the input string $w$. We use BOS and EOS symbols, distinct symbols that are placed at the beginning and end of every input string, respectively.

$\mathcal{L}^{(\ell)}$ for $\ell \in [L]$ denotes a **transformer layer**—a mapping $\mathcal{L}^{(\ell)}\colon \left(\mathbb{R}^D\right)^* \to \left(\mathbb{R}^D\right)^*$ that updates the symbol representations. The components of a transformer layer are the **layer normalization** LN, the **attention layer** $\boldsymbol{f}_{\mathsf{att}}^{(\ell)}$ and the **feedforward network** $\mathbf{F}^{(\ell)}$. Concretely:

$$\mathcal{L}^{(\ell)} \stackrel{\text{def}}{=} \mathbf{F}^{(\ell)} \circ \boldsymbol{f}_{\mathsf{att}}^{(\ell)} \circ \mathsf{LN}^{(\ell)} \tag{3}$$

---

[4]To guarantee the transformer width is constant while the number of layers grows with input length, we recall transformer layers can reset intermediate values in looping layers (Merrill & Sabharwal, 2025a).

We recall layer-normalization maps a vector $\boldsymbol{x} \in \mathbb{R}^n$ of some dimension $n$ to $\frac{\boldsymbol{x}'}{\|\boldsymbol{x}'\|}$ where $\boldsymbol{x}' \stackrel{\text{def}}{=} \boldsymbol{x} - \frac{\sum_{x_i \in \boldsymbol{x}} x_i}{n}$. We assume **multi-pre-norm** (Merrill & Sabharwal, 2024). In standard pre-norm, we apply a layer-normalization to the entire hidden state of each symbol. In multi-pre-norm, we allow each sublayer to take $k$ different projections of its input apply layer-norm to each and concatenate. Crucially, multi-pre-norm allows us to partition the hidden state and normalize disjoint subsets of thereof, which we will rely on in our proofs.

$\mathbf{F}^{(\ell)} \colon \left(\mathbb{R}^D\right)^* \to \left(\mathbb{R}^D\right)^*$ is a position-wise function that applies the same feedforward network to every symbol of the sequence. It is parametrized by weight matrices of the form $\boldsymbol{W} \in \mathbb{R}^{m \times D}$ and $\boldsymbol{U} \in \mathbb{R}^{D \times m}$. A feedforward network $\mathbf{F}^{(\ell)}$ can nest functions of the form $\boldsymbol{U}\mathrm{ReLU}(\boldsymbol{W}\boldsymbol{z})$ where $\boldsymbol{z} \in \mathbb{R}^D$ is an intermediate value.

The **attention mechanism** is defined by the function $\boldsymbol{f}_{\mathsf{att}}^{(\ell)} \colon \left(\mathbb{R}^D\right)^* \to \left(\mathbb{R}^D\right)^*$. We denote by $\boldsymbol{k}_i^{(\ell)}$, $\boldsymbol{q}_i^{(\ell)}, \boldsymbol{v}_i^{(\ell)}$ the key, query and value vectors, respectively, for symbol $i$ at layer $\ell$. $\boldsymbol{f}_{\mathsf{att}}^{(\ell)}$ is defined as follows:

$$\boldsymbol{f}_{\mathsf{att}}^{(\ell)}((x_1, \cdots, x_T)) \stackrel{\text{def}}{=} (y_1, \cdots, y_T) \tag{4a}$$

$$y_i \stackrel{\text{def}}{=} x_i + \sum_{i' \in m(i)} s_{i'} \boldsymbol{v}_{i'}^{(\ell)} \tag{4b}$$

$$s = \mathsf{proj}(\{\mathsf{score}(\boldsymbol{k}_{i'}^{(\ell)}, \boldsymbol{q}_i^{(\ell)})\}) \tag{4c}$$

$m(i)$ is a set that defines the **masking** used by the transformer. For instance, $m(i) = \{i' \mid i' < i\}$ refers to strict causal masking and $m(i) = [\|\boldsymbol{w}\|]$ refers to no masking. $\mathsf{score}$ is a scoring function that maps two vectors of the same size to a scalar. Typically, the dot-product score is used with $\mathsf{score}(x_1, x_2) \stackrel{\text{def}}{=} \langle x_1, x_2 \rangle$.

Throughout layers, the hidden state $y_i$ of a symbol at position $i$ continuously evolves as it cumulatively adds up the outputs of the attention mechanism. We call this cumulative sum $y_i$ over layers the **residual stream** at $i$.

$\mathsf{proj}$ is a projection function that normalizes the scores into weights for the symbol values. Following previous work, we assume an **averaging hard attention** transformer (AHAT), which concentrates the attention weights on the symbols that maximize the attention score (Merrill et al., 2022; Strobl, 2023). Formally, we have $\mathsf{proj} = \mathrm{hardmax}$:

**Definition A.1.** *Averaging hard attention is computed with the* $\mathrm{hardmax}$ *projection function:*

$$\mathrm{hardmax}\left(\boldsymbol{x}\right)_d \stackrel{\text{def}}{=} \begin{cases} \frac{1}{m} & \textbf{if } d \in \mathrm{argmax}\left(\boldsymbol{x}\right) \\ 0 & \textbf{otherwise} \end{cases} \tag{5}$$

*for $d \in [D]$, where $\boldsymbol{x} \in \mathbb{R}^D$ and $m \stackrel{\text{def}}{=} |\mathrm{argmax}\left(\boldsymbol{x}\right)|$ is the cardinality of the argmax set.*

**Recognition.** A transformer is a vector-valued function. To link this to language recognition, we use the representations computed by a transformer for binary classification of strings. We denote by $\boldsymbol{x}_{\mathrm{EOS}}^L$ the hidden state of EOS at the end of the forward pass of $\mathsf{T}$. Typically, string recognition is based on $\boldsymbol{x}_{\mathrm{EOS}}^L$ as EOS is the only symbol that is able to access information about every single symbol throughout all (assuming causal masking). This allows us to define a transformer's language based on a linear classifier:

$$\mathbb{L}(\mathsf{T}) \stackrel{\text{def}}{=} \{\boldsymbol{w} \in \Sigma^* \mid \boldsymbol{\theta}^\top \boldsymbol{x}_{\mathrm{EOS}}^L > 0\}. \tag{6}$$

**Precision.** Following previous work (Merrill & Sabharwal, 2025b; 2024; 2023), we assume log-precision transformers, i.e., we allow the transformer to manipulate values that can be represented with $\mathcal{O}(\log(n))$ bits for an input of length $n$. It is a minimally extended idealization that enables the transformer to store indices and perform sums over an unbounded number of symbols, two crucial capabilities for our constructions.

### A.1.2 LAYER-NORM HASH

We will often use the **layer-norm hash** building block (Merrill & Sabharwal, 2024). It is particularly useful for equality checks between values across different symbols, especially with a potentially unbounded number of queries and keys.

**Definition A.2** (Merrill & Sabharwal, 2024). *Given a scalar $z \in \mathbb{R}$, its **layer-norm hash** is $\phi(z) \stackrel{\text{def}}{=} \langle z, 1, -z, -1 \rangle / \sqrt{z^2 + 1}$.*

Layer-norm hash is scale invariant, and $\phi(q) \cdot \phi(k) = 1$ if and only if $q = k$. In other words, the inner product of scalars $q$ and $k$, even if computed at different positions $i$ and $j$, respectively, allows us to check for the equality of $q$ and $k$. Layer-norm hash therefore allows us to perform equality checks over elements of residual streams at different positions.

# B  TRANSFORMER CONSTRUCTIONS PROOFS

In our constructions, we leverage padding tokens to associate them with distinct objects. For example, when computing the realizability of items in Alg. 1 and Alg. 2 on AHAT$s$, we will associate each item with a padding token. To this extent, we introduce a novel theoretical gadget implementable by AHAT$s$ that enables a padding token at some position $i$ to compute the encoding of its associated items from the unique position $i$. We formalize this statement in the following lemma:

**Lemma B.1** (Converting a padding token position into a binary representation). *Let $\mathsf{T}$ be a $\mathcal{O}(\mathcal{P}(n))$-padded transformer. Let $\mathcal{S} = \mathcal{S}_1 \times \mathcal{S}_1 \ldots \mathcal{S}_m$ be some set such that its elements can be represented with $\mathcal{O}(\log(\mathcal{P}(n)))$ bits. Then, in a constant number of layers, each padding token can add to their residual stream the encoding of a distinct element of $\mathcal{S}$.*

*Proof.* Firstly, a padding token at position $i$ can add to the residual stream $\phi(i)$ with one causally-masked attention layer by uniformly attending over the strict left context and setting as value $\mathbb{1}[i = 0]$(Merrill & Sabharwal, 2024).

Each padding token is distinguished by its unique position. We will rely on this fact to unpack bits of the binary representation of $\phi(i)$ to store the encoding of a *distinct* element of $\mathcal{S}$.

Recall AHATs can compute Euclidean divisions and modulo at some position $i$ for integers smaller than $i$ in a constant number of layers (Merrill & Sabharwal, 2025a). We leverage this theoretical gadget to partition the binary representation of $\phi(i)$ into an element of $\mathcal{S} = \mathcal{S}_1 \times \mathcal{S}_1 \ldots \mathcal{S}_m$. As an example, suppose $\mathcal{S}_1 = [n]$, and $s_1$ is some index in $\mathcal{S}_1$. $s_1$ can then be written with $\log(n)$ bits. We can *extract* $s_1$ from $\phi(i)$ by considering the binary representation of the latter and extracting the first $\log(n)$ bits or equivalently, computing $\phi(i)$ MOD $n$. To add to the residual stream the next element $s_2 \in \mathcal{S}_2$, we can clear out the first $\log(n)$ bits of $\phi(i)$ by dividing $\phi(i)$ by $n$. This example illustrates how we can extract from $\phi(i)$ an element of $\mathcal{S}$: we iteratively 1) mask the first $\log(|\mathcal{S}_i|)$ bits from the least significant bit to extract an element of $\mathcal{S}_i$ and 2) shift the binary representation of $\phi(i)$ towards the least significant bit to then extract the following element in $\mathcal{S}_{i+1}$. ∎

## B.1  GENERAL CFL RECOGNITION ON TRANSFORMERS

**Theorem 3.1.** *Given a CFL $\mathbb{L}$, there exists a transformer with both causally-masked and non-masked attention layers, $\mathcal{O}(\log(n))$ looping layers and $\mathcal{O}(n^6)$ padding tokens that recognizes $\mathbb{L}$. That is, CFL $\subseteq$ mAHAT$_6^1 \subseteq$ AHAT$_7^1$.*

*Proof.* We store padding tokens for each possible item (of the form $[X, i, j]$ or $[X, i, j]/[Y, k, l]$) and each possible way to decompose that item. There are $\mathcal{O}(n^6)$ such tokens: In the worst case, we are solving an item $[X, i, j]/[Y, k, l]$ and are guessing an item $[Z, p, q]$ that decomposes that problem. Intuitively, if a padding token aims to solve the item $[X, i, j]$ and holds as decomposition $[Y, k, l]$, we attend to the padding tokens which solve $[X, i, j]/[Y, k, l]$ and $[Y, k, l]$. Due to Thm. 3.3, if $[S, 1, n]$ is realizable then there exists a padding token with associated item $[S, 1, n]$ such that it will store 1 (denoting realizability) in its residual stream after $\mathcal{O}(\log(n))$ steps.

We firstly detail how each padding token can add to their residual stream the encodings of their associated item and subsequent decomposition. A padding token at position $i$ can add to their residual stream $\phi(i)$ with one causally-masked attention layer by attending to their strict left context (Merrill & Sabharwal, 2024). We define the set $\mathcal{S} = \mathcal{S}_1 \times \ldots \mathcal{S}_m$ as the set of all possible item / decomposition combinations. For instance, $([X, i, j], [Y, k, l])$ is an element of this set, where we will decompose $[X, i, j]$ into $[X, i, j]/[Y, k, l]$ and $[Y, k, l]$. $\mathcal{S}_1$ could contain a set of non-terminals in $\mathcal{N}$, $\mathcal{S}_2$ could contain a set of indices in $[n]$, so on and so forth. Finally, we leverage Lem. B.1 to add the encodings

of these elements in the residual stream. For each padding token we can therefore store its associated item and decomposition.

We will now detail how to compute the realizability of items associated with these padding tokens. We consider items of the form $[X, i, j]$, solving items of the form $[X, i, j]/[Y, k, l]$ follows the same idea.

Padding tokens allocate space for an element of $\{0, 1, \perp\}$, which describes whether the associated item is non-realizable (0), realizable (1), or not known yet to be realizable ($\perp$). Padding tokens initially all store $\perp$.

**Base case:**  Items of the form $[X, i, j]$ are a base case item if $i = j$. A feedforward network can for each padding token associated with some $[X, i, j]$ check that $i = j$ by adding $i - j$ to the residual stream. With an attention layer, we can then retrieve and add to the residual stream the encoding of the symbol $w_i$ for a given base case item $[X, i, i]$ as follows. A symbol representation at position $i$ can add to its residual stream $\phi(i)$ by uniformly attending with a causally-masked attention layer to all symbol representations in the strict left context (Merrill & Sabharwal, 2024). A padding token associated with $[X, i, i]$ also stores $\phi(i)$. Therefore, via an equality-check via dot product, padding tokens can attend to relevant symbol representations by setting as value the one-hot encoding of the symbol $[\![w_i]\!]$. Finally, a feedforward network can add to the residual stream 1 if $X \to w_i$ is a valid rule and otherwise 0: A mapping between two finite sets $\mathcal{N} \times \Sigma \to \{0, 1\}$ can be computed by a feedforward network.

**Induction step:**  Recall a padding token stores 1) an item to solve (for instance, $[X, i, j]$) and 2) a set of objects that enable us to decompose that item (for instance, $[Y, k, l]$). Given $[X, i, j]$, $[Y, k, l]$, a feedforward network adds the encodings of $[X, i, j]/[Y, k, l]$ and $[Y, k, l]$ to the residual stream. Otherwise, if a padding token is associated with $[X, i, j]$, $X \to YZ$ and $k$, we add $[Y, i, k - 1]$ and $[Z, k, j]$ to the residual stream via a feedforward network. In the latter case, a feedforward network can also ensure the rule $X \to YZ$ is in the grammar, and store 0 in the residual stream (denoting non-realizability) if the rule is not in the grammar.

Finally, with one attention layer and a feedforward network, we can attend to all padding tokens that aim to solve the first subproblem ($[X, i, j]/[Y, k, l]$) and copy the integer in the allocated cell for realizability. We also perform the same procedure for the second subproblem to solve.

We compute the realizability of the current item via an extension of standard Boolean logic (Tab. 3) to handle the case where padding tokens have not yet computed the realizability of their associated item. We do not elicit the standard rules of propositional logic for brevity. Crucially, a feedforward

| $P$ | $Q$ | $P \wedge Q$ | $P \vee Q$ |
|---|---|---|---|
| 1 | $\perp$ | $\perp$ | 1 |
| $\perp$ | 1 | $\perp$ | 1 |
| 0 | $\perp$ | 0 | $\perp$ |
| $\perp$ | 0 | 0 | $\perp$ |
| $\perp$ | $\perp$ | $\perp$ | $\perp$ |

Table 3: Truth table for a three-valued logic
that handles propositions with unknown truth value.

network can compute this mapping as it is between two finite sets.

After at most $\log(n)$ steps, some padding token aiming to solve an item $[A, i, j]$ will necessarily store 1 if and only if $[A, i, j]$ is realizable: There exists some balanced decomposition represented by two padding tokens that we can attend to and store the realizability of their associated items.

**Recognition step:**  The EOS token can uniformly attend to all padding tokens that encode the item $[S, 1, n]$ (we can add $S, 1$ and $n$ to the residual stream beforehand) item and ensure one of them holds 1, denoting realizability. ∎

## B.2 Unambiguous CFL recognition on transformers

**Lemma 4.1.** *Let $\psi$ be a variable-free Boolean formula. Assume $\psi$ is represented in a transformer's residual stream as follows, where we consider the binary tree induced by $\psi$. For each leaf, there is a padding token that encodes its value (1 or 0). For each function node, there is a padding token that encodes its type ($\wedge$ or $\vee$) and pointers to its input arguments. Then, we can compute the value of each subformula in $\mathcal{O}(\log(n))$ time on an input of length $n$.*

*Proof.* We will implement Rytter (1985)'s parallel pebble game algorithm for evaluating Boolean formulas in $\mathcal{O}(\log(n))$ steps. We first formalize different objects we associate with a node. Recall every vertex $v$ in the binary tree induced by $\psi$ is represented by some padding token which stores pointers to its input arguments. For the padding token associated with vertex $v$, we allocate space for the following objects:

- VALUE is the result of evaluating the formula associated with $v$.

- PTR is a pointer to a vertex in the computation tree. Initially, all padding tokens store a pointer to themselves. Intuitively, if the value of PTR is known, we can compute the value of the formula associated with $v$.

- CONDF: $\{0, 1\} \rightarrow \{0, 1\}$ is a conditional function that relates PTR's value to $v$'s value with $v.\text{VALUE} = \text{CONDF}(\text{PTR.VALUE})$.

The parallel pebbling game consists of three steps which are repeated $\mathcal{O}(\log(n))$ times: `activate`, `square` and `pebble`. We introduce each operation and detail how to perform them on AHATs.

`activate`: Recall that $v$'s padding token stores pointers to its input arguments $v_1$ and $v_2$. If the value of $v_1$ is known, PTR is set to $v_2$ (and vice-versa). $v$'s padding token can attend to $v_1$'s and $v_2$'s padding tokens via an equality-check and copy $v_1.\text{VALUE}$ and $v_2.\text{VALUE}$. Suppose that $v_1$'s value is known (the symmetric argument with $v_2$ is the same). We will detail how to define $v$'s CONDF depending on $v_1$'s value and $v$'s function type. For instance, if $v$'s function type is $\wedge$ and $v_1$ is known to evaluate to 1, we know $v$'s value is exactly PTR.VALUE, and therefore we define the conditional function as $\text{CONDF}(x) = x \; \forall x \in \{0, 1\}$. We detail all the distinct cases in the following table.

| $v$'s function type | $v_1.\text{VALUE}$ | conditional function type |
|:---:|:---:|:---:|
| $\vee$ | 1 | $\text{CONDF}(x) = 1 \; \forall x \in \{0, 1\}$ |
| $\vee$ | 0 | $\text{CONDF}(x) = x \; \forall x \in \{0, 1\}$ |
| $\wedge$ | 1 | $\text{CONDF}(x) = x \; \forall x \in \{0, 1\}$ |
| $\wedge$ | 0 | $\text{CONDF}(x) = 0 \; \forall x \in \{0, 1\}$ |

Table 4: Defining $v$'s relation to PTR's value depending on $v_1.\text{VALUE}$ and $v$'s function type.

Feedforward networks are able to compute conditional functions (Yang et al., 2025). Therefore, a feedforward network can add to $v$'s residual stream a pointer to PTR, 0 or 1 depending on the cases presented in App. B.2.

`square`: We then compute the one-step closure of ACTIVATE. Let $v.\text{PTR} = v'$ and $v'.\text{PTR} = v''$. We first update $v.\text{PTR}$ with $v'.\text{PTR} = v''$ by having $v$'s padding token attend to $v'$'s padding token and copy $v'.\text{PTR}$. Furthermore, by copying $v''$'s CONDF via another attention layer, a feedforward network can compose the conditional functions of $v$ and $v'$.

`pebble`: Finally, we evaluate $v.\text{VALUE}$ at the current iteration by setting $v.\text{VALUE}$ $=\text{CONDF}(v.\text{PTR.VALUE})$ via another feedforward network.

We refer to Rytter (1985) for the original presentation of this algorithm and the proof of the $\mathcal{O}(\log(n))$ time bound. ∎

**Theorem 4.1.** *Let UCFL be the classes of unambiguous CFLs. Then $\text{UCFL} \subseteq \text{mAHAT}_3^2 \subseteq \text{AHAT}_4^2$.*

*Proof.* Each item $[A, i, j]$ is associated with a padding token. Each potential edge between vertices representing items $[A, i, j], [B, i, k]$ is associated with a padding token. There are $\mathcal{O}(n^3)$ such padding tokens. We leverage Lem. B.1 to enable padding tokens to add to their residual stream the encodings of their associated items from $\phi(i)$, the layer-norm hash of their position $i$.

Each padding token for vertices allocates space to store an element in $\{0, 1, \perp\}$ to denote that the associated item is either non-realizable (0), realizable (1) or not known yet to be realizable ($\perp$). We will implement Alg. 3's algorithm on AHATs to compute whether items are part of the closure $\mathcal{C}(\boldsymbol{w})$ (i.e, are realizable) or not.

**Initial items:** A padding token for some vertex can check whether its associated item is of the form $[A, i, i]$ via a feedforward network that checks that the indices are the same. For all such padding tokens, another feedforward network adds 1 to the residual stream if and only if $A \to w_i \in \mathcal{P}$ to signal the realizability of that item (and otherwise adds 0). We can perform this procedure exactly as in the base case of App. B.1.

**Creating the dependency graph:** Padding tokens for edges store items of the form $[A, i, j], [B, i, k]$. There are finitely many $[C, k+1, j]$ such that $A \to BC \in \mathcal{P}$ (proportionally many in $|\mathcal{N}|$), which can be added to the residual stream via a feedforward network. According to Eq. (1), we set an edge between vertices associated with $[A, i, j]$ and $[B, i, k]$ if and only if there is an item $[C, k+1, j]$ such that $[C, k+1, j]$ is realizable (i.e, the corresponding padding token stores 1 in its residual stream) and $A \to BC \in \mathcal{P}$. The padding token for the edge associated with $[A, i, j], [B, i, k]$ can check whether any of the items of the form $[C, k+1, j]$ are realizable and satisfies $A \to BC \in \mathcal{P}$ via an equality-check with an attention layer (to check the realizability of the items) and a feedforward-network (to check whether $A \to BC \in \mathcal{P}$). If such an item exists, the padding token associated with $[A, i, j]$ and $[B, i, k]$ signals that there is an edge between them in the dependency graph.

**Binarization:** Due to unambiguity, there is at most one path between any pair of vertices in the dependency graph. If there are multiple paths from a vertex $[A, i, j]$ to another vertex $[B, k, l]$, there are then different derivations that can reduce $[A, i, j]$ to $[B, k, l]$, which contradicts the unambiguity condition. Evaluating reachability queries on a tree reduces to solving the Boolean formula induced by this tree where leaf vertices are assigned 1 or 0 depending on if they are associated with realizable items and non-leaf vertices are assigned the $\vee$ operator.

However, to efficiently evaluate this Boolean expression, we require a *binary* tree where each vertex has at most two children. For every block of looping layers, we consider the binarization of the dependency graph as follows.

The binarization assumes we have $\mathcal{O}(n^3)$ additional padding tokens appended to the input, i.e., $\mathcal{O}(n)$ extra padding tokens for every vertex for item. Note that asymptotically, appending $\mathcal{O}(n^3)$ padding tokens does not impact our claim on the resource bounds required. Effectively, for some item $[A, i, j]$, we have a $n$-arity tree and need to use $\mathcal{O}(n)$ extra vertices to create a binary tree by replacing edges with these intermediate vertices. We will create a right-branching binary tree. We denote by $h_0$ the root node, by $v_1, v_2, \ldots v_n$ the leaf vertices, and by $h_1, h_2, \ldots h_{n-2}$ the extra intermediary vertices. We build the right-branching binary tree as follows. The root vertex has edges to $v_1$ and $h_1$, and now $h_1$ recursively needs to span $v_2, v_3 \ldots v_n$. $h_1$ then has edges to $v_2$ and $h_2$, so on and so forth. More generally, for $i < n - 2$, we instantiate the edges $(h_i, v_{i+1})$ and $(h_i, h_{i+1})$. For $i = n - 2$, we instantiate the edges $(h_{n-2}, v_{n-1})$ and $(h_{n-2}, v_n)$. The resulting tree is a binary right-branching tree.

We build the corresponding binary tree on the transformer as follows. We identify and encode the vertices of the tree using **Gorn addresses** (Gorn, 1967). A Gorn address is a bitstring such that a vertex a depth $h$ in the tree is associated with a bitstring with $h$ bits. The addresses are defined recursively. The root vertex is associated with the empty bitstring $\varepsilon$. An arbitrary vertex at tree depth $h$ associated with the bitstring $b_1 b_2 \ldots b_h$ characterizes the Gorn addresses of its two children with $b_1 b_2 \ldots b_h 0$ and $b_1 b_2 \ldots b_h 1$. For instance, Fig. 1 shows a right-branching tree with the corresponding Gorn addresses.

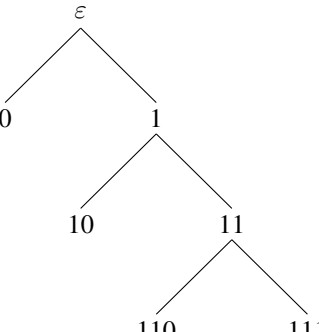

Figure 1: Right-branching binary tree with Gorn addresses as vertex labels.

Due to Lem. B.1, these padding tokens can compute their distinct Gorn address. We can assume that the padding tokens are partitioned such that the padding tokens associated with the leaves of the tree are the ones that correspond to the edges of the form $([A, i, j], [B, i, k])$. Then, the novel padding tokens not associated with items can compute the pointers to their descendants in the binary tree as follows. To compute the Gorn address of the first descendant, we shift towards the left the binary representation of the integer by multiplying it by 2 via a feedforward network. We obtain the Gorn address of the second descendant by adding 1 to the integer representation of the Gorn address of the first descendant.

**Solving reachability queries:** Reachability queries over binary trees now reduce to evaluating the Boolean formula associated with the binary tree. Leaf vertices associated with realizable items are assigned 1. A non-leaf vertex has a path to such a leaf if evaluating the induced Boolean expression where non-leaf compute $\vee$ over their children yields 1. We can therefore invoke Lem. 4.1 to evaluate this Boolean formula.

**Recognition step:** The EOS token can attend to the padding token for vertex associated with $[S, 1, n]$ and check whether it is realizable, i.e., store 1 in its residual stream.

■

## C EXPERIMENTAL SETUP

**Data.** We used Anonymous (2025)'s length-constrained sampling algorithm for CFLs to generate datasets. For D(1), D(2), Palindrome and Marked Palindrome, negative samples were either sampled at random from $\Sigma^*$ or were perturbations from positive strings. For BFVP, the negative strings were sampled Boolean formulas that evaluate to 0 as we preferred to focus on a transformer's ability to correctly evaluate a Boolean formula rather than determining if the formula is well-formed. The ability to process hierarchically nested structures is already captured by the language D($k$). The training set consists of 1 million samples with string length at most 40. The test set has 2000 samples with string length at most 80. Testing the model on strings longer than those seen in training enabled the evaluation of its ability to *generalize* out-of-distribution.

**Models and Training Procedure.** We trained causally masked looped transformers with no positional embeddings. We used the PYTORCH implementation of a transformer encoder layer with pre-norm. Following our definition of the transformer in §2.2, we instantiated our models with an initial block of 2 transformer layers, a looping block (which is repeated $\log(n)$ times or once at inference) of 2 transformer layers and a final block of 2 transformer layers. A binary classifier (2 layer feedforward network) was then applied to the final contextual representation of EOS. Our transformers have 1.2 million parameter budget. We used the ADAMW optimizer (Loshchilov & Hutter, 2019) and binary cross-entropy loss, considering runs across 5 different seeds. The batch size was set to 64 and the learning rate to 0.0001.

