# OpenReview forum: "Context-free Recognition with Transformers"
_ICLR.cc/2026/Conference — Submitted to ICLR 2026_

### Official Review · Reviewer_vZZD · 2025-10-18

**Soundness:** 3
**Presentation:** 3
**Contribution:** 3
**Rating:** 6
**Confidence:** 3

**Summary:**

This paper provides a theoretical analysis of the ability of transformers to recognize Context-Free Languages (CFLs). The main contribution is a constructive proof that transformers augmented with $O(\log n)$ looping layers and $O(n^6)$ padding tokens can recognize all CFLs. Furthermore, the paper shows that for subclasses of CFLs—namely unambiguous and unambiguous linear CFLs—the resource requirements, particularly the number of padding tokens, can be significantly reduced to $O(n^3)$ and $O(n^2)$ respectively. An empirical evaluation is included to validate the utility of looping on languages of varying complexity.

**Strengths:**

1. Theoretical Soundness: The core theoretical result—CFL recognition with log-depth looping—is a non-trivial and valuable contribution to the understanding of transformer expressivity. It directly addresses an open question in the field.

2. Nuanced Analysis: Moving beyond the general case to analyze subclasses like unambiguous CFLs is a strength. It provides a more refined perspective that aligns with practical observations (e.g., transformers struggling with ambiguity) and connects theory with relevant linguistic properties.

**Weaknesses:**

Empirical experiments do not provide a strong support for the theoretical analysis. In Table 2, the differences between Fixed-depth and $\log (n)$ looping in BFVP and BFVP ((postfix)) are small. Moreover, even if there are log-depth transformer constructions for D(2) and Marked Palindrome, $log (n)$ looping is still stronger than Fixed-depth. The above two empirical results seem to show that the theoretical analysis in the paper has little impacts on practical applications.

**Questions:**

Why is Fixed-depth more accurate than log(n) looping in Palindrome and D(1)? Could you provide a possible explanation?

---

> ### Author Response · Authors · 2025-11-21
>
> Thank you very much for taking the time to read our work and writing a constructive review. We appreciate your recognition of the importance of advancing work in transformer expressivity. We also thank you for noting our nuanced analysis on several classes of CFLs as they have some relevant scientific interest (LMs struggle to model ambiguity in practice, deterministic CFLs are used for programming language compilers) and we believe our work deepens our understanding of the linguistic properties of transformer-based LMs.
>
> > In Table 2, the differences between Fixed-depth and logn looping in BFVP and BFVP ((postfix)) are small.
>
> Regarding experiments, we agree that expressiveness does not paint the whole picture of the abilities and limitations of transformers. The ability of a fixed-depth transformer to generalize easily to held-out data on BFVP is a possibility that is not addressed in our work (and is out of scope) and can be studied, for instance, by deriving convergence bounds through some gradient-step analysis. While our theoretical results offer a preliminary view on which transformer variants can represent which grammars, we hope our experiments can motivate future work on developing a coherent theory of which types of grammars are more easily learnable by transformers. We will clarify in revisions the questions that are left open as opportunities for future work.
>
> > Moreover, even if there are log-depth transformer constructions for D(2) and Marked Palindrome, logn looping is still stronger than Fixed-depth.
>
> We assume you meant to write that there are fixed-depth solutions to these problems despite the looped transformer performing slightly better to held-out data. For D(2), we can partly explain this result via the C-RASP conjecture. C-RASP is a class of formal languages that softmax, fixed-depth transformers can provably recognize and generalize on ([1] [2]). As D(2) is provably not in C-RASP ([3]), we conjecture that if C-RASP is an accurate characterization of problems fixed-depth transformers can generalize on, it is thus not surprising that a fixed-depth solution for D(2) is less accurate than a looped, more complex solution. For Marked Palindrome, we conjecture that in fixed-depth solutions, positional information is usually depended on (as demonstrated in the following theoretical constructions for Palindrome [4] [5]). Particularly, we suspect a fixed-depth transformer might learn a solution more easily if aided with the explicit position of the marker, which could explain the lower accuracy of a fixed-depth solution without PEs. In contrast to our work, [6]’s results show an increased performance for Marked Palindrome compared to Palindrome with PE-augmented transformers. We argue this discrepancy arises because a transformer might learn a solution for Marked Palindrome more easily when the position of the marker is made explicit via PEs. In our results, the accuracy for out-of-distribution strings in Marked Palindrome has a high standard-deviation (relative to results for other languages). We hypothesize that the model’s ability to make use of the marker is very brittle when it is not augmented with positional information.
>
> > Why is Fixed-depth more accurate than log(n) looping in Palindrome and D(1)? Could you provide a possible explanation?
>
> The fixed-depth accuracies for D(1) for fixed depth (94%) and log depth (93%) are within the margin of error based on the standard deviations (1-2%), so we actually view them as being comparable in performance. This supports the prediction of the C-RASP conjecture that constant-depth transformers should be able to length-generalize on D(1) ([1]). Precisely, the C-RASP conjecture predicts that fixed-depth transformers should generalize better on D(1) than on D(2). which is supported by 94% accuracy on D(1) vs 83% on D(2). Interestingly, looping seems to boost the capability of transformers to generalize on D(2), lifting generalization accuracy to 90%.
>
> On the other hand, for Palindrome, it is the case that the fixed-depth transformer (79%) outperforms the log-depth transformer (72%), though neither accuracy is particularly high. Palindrome is not representable in C-RASP but it does have a fixed-depth transformer construction, which explains why generalization accuracy is lower across the board. We are not sure why looping hurts generalization here.
>
> (continued in next comment)

---

> > ### Comment · Reviewer_vZZD · 2025-11-21
> >
> > Thanks for the reply. I have increased my confidence score to 4. I believe your paper should be accepted.

---

> ### Author Response · Authors · 2025-11-21
>
> > The above two empirical results seem to show that the theoretical analysis in the paper has little impacts on practical applications.
>
> Our theoretical characterizations of expressiveness are not tight and do not predict what is learnable. They cannot fully predict the empirical results, but they do make some testable predictions confirmed in the experiments, such as looped transformers performing better than fixed-depth transformers on a log-depth complete language. Moreover, we believe they constitute a first step towards a more complete theory on the ability of transformers to internally process syntax.
>
> [1] Counting like transformers: compiling temporal counting logic Into softmax transformers, Yang et al 2024
>
> [2] A formal framework for understanding length generalization in transformers, Huang et al 2025
>
> [3] Between circuits and Chomsky: pre-pretraining on formal languages imparts linguistic biases, Hu et al 2025
>
> [4] Formal language recognition by hard attention transformers: perspectives from circuit complexity, Hao et al 2022
>
> [5] The role of logic and automata in understanding transformers, Lin et al 2025
>
> [6] Training neural networks as recognizers of formal languages, Butoi et al 2025

---

### Official Review · Reviewer_YVFz · 2025-10-20

**Soundness:** 2
**Presentation:** 2
**Contribution:** 2
**Rating:** 4
**Confidence:** 3

**Summary:**

The paper shows that context-free languages are recognizable by O(log n)-looped AHAT transformers with O(n^6)-padded tokens (this means that apart from a constant number of initial and terminal layer, we have O(log n)-time repetition of a constant block of layers). There are also improvements of the number of padded tokens for subclasses of context-free languages.

The context is that O(log n)-looped AHAT transformers coincide with the class of logspace-uniform O(log n)-depth threshold circuits (Merril, Sabharwal, 2025), and this class is subsumed by NC^2. Due to results of Ruzzo, CFL are in NC^2.

**Strengths:**

The circuit and computational complexity of CFL is an interesting topic, and transformers recently have established useful connections with circuit complexity. It is now known whether CFL is in NL, it seems unlikely that they are in NC^1.

**Weaknesses:**

The proof of the main result is a bit sloppy, and I could not understand the details (and could not restore them myself). The idea in short is that there are some guessing rules that allow to check in O(log n) recursive steps whether a given non-terminal derives a word of length n. More precisely, there are polyhnomially many ``derivation statements'' (``items'' and ``slashed items''). Each derivation statement is either an easily checkable base case or can be derived from 2 other derivation statements. Corollary 3.1 is the main non-trivial part that supposed to show that for every statement there exists an O(log n)-depth derivation. Then it is easy to see that a looped transformer can compute all derivable statements if it has enough (poly n) tokens to store information for all statements and try all possible 1-step derivations for O(log n) iterations.

The part after Corollary 3.1 is written in some detail in appendix, but it seems straighfordard anyway. The non-trivial part (Corollary 3.1) is written in the hand-waiving manner,  talking about some balanced decompositions of some trees that have not been defined, and it is not clear where all the details of the recursive cases in Lemmas 3.1 and 3.2 play the role (with the proof of Lemma 3.2 omitted).

It is not clear how much the non-transformer part is original in the paper. I asked the authors to clarify this. My current guess is that Ruzzo probably should have been doing something similar to Corollary 3.1 to obtain that CFL are in NC^2, and probably the current paper replaces the arugment that this guessing system can be simulated in NC^2 by the argument that it can be simulated in a looped AHAT.

So far, due the lack of clarity and non-evident originality, I vote for weak reject.

**Questions:**

Is some analoge of Corollary 3.1 has been established in the literature (by Ruzzo or somebody else?) If not, what new ingredients the current proof has to obtain that CFL are in log(n)-looped AHAT?

---

> ### Author Response · Authors · 2025-11-21
>
> Thank you very much for taking the time to read our work and writing a constructive review. We appreciate your recognition of the importance of understanding CFL recognition with parallel models of computation, and agree that the interplay between circuits and transformers motivates studying CFL recognition on transformers through parallel algorithms.
>
> > The proof of the main result is a bit sloppy, and I could not understand the details
>
> We have improved the writing for the main construction (namely, simulating the algorithm for CFL recognition on AHATs) and have clarified the details you brought up, and in particular, the balanced decomposition algorithm (Corollary 3.1). We now elaborate further on this.
>
> > The non-trivial part (Corollary 3.1) is written in the hand-waiving manner, talking about some balanced decompositions of some trees that have not been defined, and it is not clear where all the details of the recursive cases in Lemmas 3.1 and 3.2 play the role.
>
> To clear up your concerns, we will illustrate the correctness of the balanced decomposition algorithm. A parse tree is an object that is well-defined: a set of derivation rules applied to a non-terminal A that leads to some string w_i … w_j. Suppose we aim to determine whether A derives w_i … w_j via some parse tree. Firstly, because we consider general CFGs, many such trees may exist. However, if A derives w_i … w_j, there exists such a tree. Moreover, due to Jordan’s theorem, there exists a split node that decomposes that parse tree into two subproblems represented by trees of roughly equal size. Therefore, our algorithm leverages parallelism to check all possible decompositions of the item [A, i, j], across arbitrarily many possible parse trees, all of which are well-defined.
>
> The two recursive cases of solving an item [A,i,j] (or some slashed item) arise because we distinguish two ways to split a parse tree: we either split at the root or we do not. If we split at the root A, this implies that we need to find a rule A -> B C such that B and C derive disjoint, consecutive substrings. On the other hand, if we do not split at the root, we select some other node (e.g., a non-terminal B) of the parse tree to split it into a slashed item rooted at A and the incurred subtree rooted at B. Crucially, leveraging parallelism, we can assume there exists a padding token that deals with the decomposition such that the incurred subproblems have roughly equal size (by Jordan’s theorem).
> We thank you for reading carefully and will revise the manuscript to clarify these details.
>
> > with the proof of Lemma 3.2 omitted
>
> The proofs of Lemma 3.1 and 3.2 follow the same idea. The only distinction is about whether the leaves of the tree are associated with the string w_i … w_j (an item) or some sentential form  w_i … w_k B w_l … w_j (a slashed item). In both cases, we leverage Jordan’s decomposition theorem to decompose that tree in exactly the same manner. Precisely, in the proof of Lemma 3.1, we distinguish between splitting that tree at the root A or some other non-root node. We do exactly the same for the decomposition of a slashed item, where we instead inquire about trees where the leaves are of the form w_i … w_k B w_l … w_j.
> We will revise the manuscript so it is clear that Lemma 3.2 follows the same proof as Lemma 3.1.
>
> (continued in next comment)

---

> ### Author Response · Authors · 2025-11-21
>
> > It is not clear how much the non-transformer part is original in the paper. I asked the authors to clarify this. My current guess is that Ruzzo probably should have been doing something similar to Corollary 3.1 to obtain that CFL are in NC^2, and probably the current paper replaces the argument that this guessing system can be simulated in NC^2 by the argument that it can be simulated in a looped AHAT.
>
> > Is some analogue of Corollary 3.1 has been established in the literature (by Ruzzo or somebody else?) If not, what new ingredients the current proof has to obtain that CFL are in log(n)-looped AHAT?
>
> We address both of these concerns in this section.
>
> Our work indeed builds heavily on the existing theory of parallel algorithms for CFL recognition to better understand transformers’ CFL recognition capabilities. These algorithms were developed for various models of computation besides transformers; our results showing that they can be attained on transformers are novel.
>
> Moreover, the classical theory of parallel CFL recognition was not widely known in ML or NLP, so we believe that bringing it to bear on transformers’ capabilities is a valuable contribution (similar to how Krohn-Rhodes theory of parallel regular language recognition was connected to transformers in this top 5% paper https://openreview.net/forum?id=De4FYqjFueZ at ICLR 2023). We emphasize that much of the prior literature was scattered and messy, and we believe our synthesis goes a long way towards making it accessible to the modern ML community. For instance, Ruzzo doesn’t have a theorem corresponding directly to Corollary 3.1, but instead obtains the result that CFLs are in NC2 indirectly by showing NC2 can simulate alternating Turing machines with polynomial tree size [1]. The result that CFL recognition is in AC1 (i.e., log-depth, unbounded fan-in circuits) follows implicitly from Ruzzo’s constructions (as hinted in [2] and [3]). After understanding the original proof technique, we took great effort to simplify it to be more constructive, enabling us to show how transformers could implement CFL recognition. As transformers can be viewed as parallel models of computation with unbounded fan-in ([4] [5]), we leverage these findings to show how log-depth transformers can recognize all CFLs. Crucially, the guessing system requires unbounded fan-in to be able to guess over an unbounded number of objects (w.r.t. string length n), which is why the nuance between AC1 and NC2 is important in this context.
>
> [1] Tree-sized bounded alternation, Ruzzo 1980
>
> [2] Properties that characterize LOGCFL, Venkateswaran 1990
>
> [3] A compendium of problems complete for P, Greenlaw et al 1991
>
> [4] Saturated transformers are constant-depth threshold circuits, Merrill et al 2022
>
> [5] Exact expressive power of transformers with padding, Merrill et al 2025

---

### Official Review · Reviewer_8ij9 · 2025-10-28

**Soundness:** 3
**Presentation:** 4
**Contribution:** 3
**Rating:** 6
**Confidence:** 4

**Summary:**

This paper investigates the context-free language (CFL) recognition capabilities of Transformer models. The authors present a model of Transformers with looped layers and padding tokens. They give an upper bound on the number of padding tokens ($O(n^6)$) and the number of looped layers ($O(\log n)$) required. They discuss how for simplified versions of CFL languages such as unambiguous CFLs (with one derivation tree per valid word), one can obtain models with smaller size/less padding tokens.

The authors validate their claims empirically by training transformer models to recognize different CFLs such as Palindromic languages, Dyck languages and Boolean formula value problems. Their empirical results show that transformers can indeed learn to recognize context free languages with a high degree of accuracy. Moreover, their analysis shows that using looped layers can increase accuracy.

I have given this paper a 6, but if the authors include an Experimental Details section in the appendix, I will raise my score to an 8.

**Strengths:**

* Paper is very well written. I found the theoretical exposition to ber quite clear. The approach taken by the authors to construct the algorithms and conduct their analysis before discussing relation/implementation with Transformers was very clear and pleasant to read.
* The theoretical results are novel and interesting. As the authors have also stated, this work is the first theoretical contribution (to the best of my knowledge as well) which proves CFL recognition with transformers.
* The theoretical approach is non-trivial and a significant contribution to current research on formal languages and deep learning.
	* The first proof introduces a novel approach (different than CKY) to find a valid derivation tree for a candidate word.
	* The application of path system results to Transformer expressivity (For the proof in the UCFLs case) is also (to the best of my knowledge) a novel contribution
* The analysis of CFLs with different levels of constraints also adds depth to the paper. I find the analysis of the tradeoffs here to be very interesting.
* Assumptions about the model are clear.
* Proofs seem correct and are written in a style which is easy to follow.

**Weaknesses:**

- The biggest weakness for me is experiments section. I feel alot of methodological details are missing from this section. See the the "Questions" section below. I feel an appendix section with more experimental details could be beneficial to readers who want to better understand/reproduce similar setups.
- I find the model of Transformers presented by the authors to be unrealistic and quite detached from practice. I have no problem with the use of hard attention however:
	- Padding tokens are unrealistic; no one really uses this in practice to augment model performance
	- Looping layers are also (to the best of my knowledge) not standard practice
	- Could the authors explain how such concepts can be tied to methods of practical relevance? (Maybe here they could also put emphasis on how their approach could inform future model design instead?)
- Although the proofs are easy to read. My only minor complaint is that they are very text heavy. For instance, I feel the discussion of embeddings used in the proof of Theorem 3.3 could have been clearer if simply presented in vector notation.

**Questions:**

**Theoretical Results**
* What does guessing mean in Algos 1 and 2? sampling uniformly at random?
* What is the "width" of the transformer needed for Theorems 3.3 and 4.1? Reading the proofs it seems like they would be constant w.r.t. the input length. Could this information be added somewhere? For instance in the Theorem definitions given in the appendix.
* "These results imply that, in order to recognize CFLs, transformers require significantly less depth than that which would be needed to implement a serial parsing algorithm like CKY" Can you comment on the order of magnitude (as a function of $n$) needed for a model to implement CKY?

**Experimental Results**
* How many samples were used for training?
* How were the samples generated? Were the models trained on a fixed dataset for some number of epochs or were batches sampled at training time?
* What is the maximum length of strings models are trained on?
* What are the  lengths of strings in the OOD case?
* How are looping layers implemented in practice?
* Do the authors have a hypothesis as to why the difference between using and not using looped layers is so small? I would have expected a bigger margin here given the theoretical results.

**General Questions/Remarks**
* Can the authors comment on how augmenting a Transformer model with a CoT would change the size needed? Do the authors have a sense of how many CoT steps would be needed to recognize a word in a CFL?
* Do the authors see any connections between looped layers and continuous CoT frameworks?
* You cite the Zhao et al. paper "Do Transformers parse while predicting the masked word?", but do not discuss the theoretical results of this paper nor its implication of this to your own theoretical results. Would it be possible to comment on how the approach/results in your work differ/contrast to those in the work of Zhao et al.
* It could be interesting to have interpretability results as well. I am curious to see what kind of solutions models learn through gradient-based training.

**Minor Comments and Typos**
* Is there in error in the production rule definition (3) at line ~092? We would need $\alpha \in (\mathcal{N} \cup \Sigma)^*$.
* line ~162: "exists a rule $X \to YZ$ and index $k$ s.t. $[Y, i, k-1]$ and $[X,k,j]$ are realizable" shouldn't it be $Z$ instead of $X$ here?
*  Line 447 "Our transformers has 1.2 million parameter budget" typo here

---

> ### Author Response · Authors · 2025-11-21
>
> Thank you very much for taking the time to read our work and writing a detailed constructive review, we appreciate your strong implication in reviewing our work! We appreciate your recognition of the value of understanding theoretically how transformers can process CFGs. Particularly, we believe a contribution of our work is synthesizing foundational algorithms for parallel recognition of CFLs to translate them to transformers, and thank you for acknowledging these efforts.
>
> Regarding your positive comment on the application of path systems to transformers, we would also like to note that a very recent line of work ([1])  examines the complexity of a mathematical proof as the shortest path from a leaf node to a target node in a path system and how transformers can efficiently find such a proof. We find this formalization interesting and it indirectly relates to our implementation of path systems on AHATs for UCFL recognition. Future work investigating the ability of transformers to operate on path systems may therefore benefit from our work.
>
> We now partition our responses below into the different topics from your review (Weaknesses, Theoretical results, Experimental results,  General questions).

---

> ### Author Response · Authors · 2025-11-21
> **Weaknesses**
>
> > The biggest weakness for me is experiments section. I feel alot of methodological details are missing from this section. See the the "Questions" section below. I feel an appendix section with more experimental details could be beneficial to readers who want to better understand/reproduce similar setups.
>
> We have included an Experimental Setup section in the appendix, and we agree it clarifies the details of the setup behind our experiments. We also provide the answers to your questions about the experimental setup below in the Experimental Results section. We hope we have addressed this concern.
>
> > Padding tokens are unrealistic; no one really uses this in practice to augment model performance. Looping layers are also (to the best of my knowledge) not standard practice.
>
> > Could the authors explain how such concepts can be tied to methods of practical relevance? (Maybe here they could also put emphasis on how their approach could inform future model design instead?)
>
> We acknowledge that looping layers and padding tokens are not standard practice, but believe that theory-focused works (such as ours) should strive to improve the models used in practice by exploring novel extensions and methods. Looping layers and padding tokens offer a promising view on how to extend the current expressive power of transformers ([4]). Looped transformers have seen success at reasoning ([10]) and length generalization ([5]). Padding tokens have demonstrated empirical success at reasoning tasks in several works ([6] [7] [8]). Research on looped- and padded-transformers can guide novel architectures of practical relevance: recently, they have been linked with masked diffusion language models ([11]).
> Overall, we believe that dynamically-scaling transformers (with looping and/or padding tokens) could guide future work on developing better reasoning models. The time- and space-complexity of an algorithm is measured w.r.t. to input length, and classical models of computation on which these algorithms are usually implemented in theory (e.g, Boolean circuits) are defined as well w.r.t. input length. We believe dynamically-scaling transformers (w.r.t. input length) offer a better solution for learning general algorithms and as an example our work demonstrates theoretically how such a model can parse any string given some CFG.
>
> > Although the proofs are easy to read. My only minor complaint is that they are very text heavy. For instance, I feel the discussion of embeddings used in the proof of Theorem 3.3 could have been clearer if simply presented in vector notation.
>
> Thank you for engaging closely with the theoretical details! We take your point that Theorem 3.3 was overly text heavy and have made revisions to address this. We have isolated the theoretical gadget to leverage the layer-norm hash of the token position to store associated items in its own lemma (at the beginning of Section B in the appendix) to cut down on the amount of text in the core proofs. We hope these changes improve readability and better convey the intuition for how parallel algorithms can be run on transformers.

---

> ### Author Response · Authors · 2025-11-21
> **Theoretical results**
>
> > What does guessing mean in Algos 1 and 2? sampling uniformly at random?
>
> Thanks for pointing this out. Non-deterministic guessing is a crucial element of parallel algorithms used in our main proofs that we will make sure to better explain in the revised version of our paper. In short, guessing means using non-determinism to find a bitstring of some length (w.r.t. input length) that satisfies some desired property. Non-deterministically guessing short bitstrings can be simulated on a deterministic parallel model of computation by enumerating different strings in parallel.
>
> For example, in our constructions, an item [A,i,j] can be decomposed into an unbounded (w.r.t. input length) number of ways. On a non-parallel model of computation, we would have to iterate through all these decompositions sequentially (which is why CKY takes n^3 time, as it loops through all these decompositions to fill up the parse table). In contrast, previous work ([2]) that assumes a parallel model of computation leverages parallelism to simultaneously check these decompositions in parallel. Precisely, a common model of parallel computation are Alternating Turing Machines (ATMs): informally, an ATM state can transition to an unbounded number of subsequent states, and only one of these choices needs to lead to an accepting state for the overall run to signal acceptance. For CFL recognition, this insight is leveraged to have ATMs attend to all ways [A,i,j] can be decomposed in parallel (rather than check them one by one). The term guessing is often used informally to denote the procedure of being able to guess the correct subsequent ATM state to attend to out of an unbounded number of ways to transition ([2]). In our work, we transpose this insight to transformers: it suffices to have a single witness padding token that demonstrates that [A,i,j] is realizable. Akin to how ATMs can transition to several states non-deterministically (in parallel), tokens update their hidden states in parallel.
>
> We will make this notion more clear in the paper, as it is a key theoretical tool for showing log-depth transformers can recognize CFLs. We thank you for raising this point and hope we clarified what we mean by guessing.
> > What is the "width" of the transformer needed for Theorems 3.3 and 4.1? Reading the proofs it seems like they would be constant w.r.t. the input length. Could this information be added somewhere? For instance in the Theorem definitions given in the appendix.
>
> Indeed, our transformer construction is achievable with constant width. We will clarify this in the theorem statement.
>
> > "These results imply that, in order to recognize CFLs, transformers require significantly less depth than that which would be needed to implement a serial parsing algorithm like CKY" Can you comment on the order of magnitude (as a function of ) needed for a model to implement CKY?
>
> CKY, in contrast to our algorithm, solves every item of the form [A,i,j] via 3 nested for loops by sequentially filling in the parse table in O(n^3) iterations. Thus, it would take sequential time O(n^3). In contrast, our parallel construction has sequential runtime (depth) O(log n). This constitutes an exponential speedup in sequential runtime. We will clarify this in revisions.

---

> ### Author Response · Authors · 2025-11-21
> **Experimental results**
>
> Thank you for your suggestions about experimental details to clarify. We answer your questions below, and have also added an appendix to the PDF with this information.
>
> > How many samples were used for training?
>
>  1 000 000 strings were sampled for the training set.
>
> > How were the samples generated?
>
> The samples were generated via a length-constrained algorithm for sampling strings from some given CFG. The algorithm is from an anonymous paper, which will normally soon be available to the public.
>
> > Were the models trained on a fixed dataset for some number of epochs or were batches sampled at training time?
>
> Batches were sampled at training time.
>
> > What is the maximum length of strings models are trained on?
>
> For strings in the training set, the maximum string length is 40.
> > What are the lengths of strings in the OOD case?
>
> For OOD strings, the maximum string length is 80.
>
> > How are looping layers implemented in practice?
>
> Our transformers have 1) an initial block of 2 layers 2) a looping block of 2 layers 3) a final block of 2 layers. At inference, the looping block is looped over log(n) times.
>
> > Do the authors have a hypothesis as to why the difference between using and not using looped layers is so small? I would have expected a bigger margin here given the theoretical results.
>
> The theoretical results only inform us about expressivity and not learnability (which could constitute future work), and can therefore not fully predict the behavior we see in practice from these models. Several factors may come into play when explaining the small gap in accuracy between looped and non-looped transformers.
>
> - Most of the problems we train transformers on have a fixed-depth solution (Dyck, Palindrome…) and therefore looping should not drastically improve performance when a fixed-depth solution is already within the representational capacity of the transformer architecture.
>
> - On the other hand, the languages where we expect to see a benefit of looping based on our theory are the BFVP variants. Indeed, we find that, for both variants, looping leads to improvements in in-distribution (1-3%) and generalization (3-4%) accuracy. We will clarify the manuscript to better highlight that, while we run experiments on various types of CFLs, BFVP is the “hard” CFL where we expect looping to help.
>
> - The length of strings may play a role. Studying generalization bounds w.r.t. the maximum input length in training could explain for which lengths in testing will a looped transformer perform better. The gap in accuracy between looped and non-looped transformers could be more stark with OOD strings of longer lengths.

---

> ### Author Response · Authors · 2025-11-21
> **General Questions/Remarks**
>
> > Can the authors comment on how augmenting a Transformer model with a CoT would change the size needed? Do the authors have a sense of how many CoT steps would be needed to recognize a word in a CFL?
>
> > Do the authors see any connections between looped layers and continuous CoT frameworks?
>
> We address both remarks on comparing looping layers with CoT here.
>
> We believe our algorithm can not be implemented under a CoT framework, as it has been proven that with O(log(n)) CoT steps, transformers are upper-bounded by TC0 (~ fixed-depth circuits) [3] while our algorithm’s tightest complexity class is AC1 (~ log-depth circuits).
> Moreover, it has been shown empirically that scaling depth (in contrast to scaling CoT steps) improves performance over regular languages recognition [3].
> Intuitively, CoT is more sequential in nature than looping as it samples a token step-by-step in contrast to updating the hidden states of tokens in parallel with a dynamically-scaling number of layers.
>
> > You cite the Zhao et al. paper "Do Transformers parse while predicting the masked word?", but do not discuss the theoretical results of this paper nor its implication of this to your own theoretical results. Would it be possible to comment on how the approach/results in your work differ/contrast to those in the work of Zhao et al.
>
> The main theoretical result from that paper considers input strings of bounded length (w.r.t. number of transformer layers) in contrast to our constructions that enable parsing a string of any input length, which is the main difference between both works. We believe implementing the Inside-outside algorithm to work for any input string would also require dynamically-scaling transformers (to store results of all possible items [A,i,j] in some memory, to iterate over all spans of the summations…). Moreover, the algorithm is inherently a sequential one as it updates iteratively the current results for the items [A,i,j] via summations until convergence (similar to the EM-algorithm), which makes us believe it does not have a natural parallel implementation on transformers.
>
> We cited this work as it shows (along with [9]) that transformers implicitly store syntactic information in their contextual representations, which drove us to examine exactly how a general algorithm for CFL recognition could be implemented on transformers.
>
> > It could be interesting to have interpretability results as well. I am curious to see what kind of solutions models learn through gradient-based training.
>
> We agree with this remark and note that interpretability results have initially sparked the motivation behind our paper. A landmark paper is [9], they show via probing that transformers encode syntax trees in their words’ representation space. Interpretability work on transformers’ ability to encode syntax prompted us to study how a concrete construction for CFG recognition might look like.

---

> ### Author Response · Authors · 2025-11-21
> **Referenced papers in our replies**
>
> [1] Are language models efficient reasoners? A perspective from logic programming, Opedal et al 2025
>
> [2] Tree-sized bounded alternation, Ruzzo 1980
>
> [3] A little depth goes a long way: The expressive power of log-depth transformers, Merrill et al 2025
>
> [4] Exact expressive power of transformers with padding, Merrill et al 2025
>
> [5] Looped transformers for length generalization, Fan et al 2025
>
> [6] Let’s think dot by dot: hidden computation in transformer language models, Merrill et al 2024
>
> [7] Learning to insert [PAUSE] tokens for better reasoning, Kim et al 2025
>
> [8] Think before you speak: training language models with pause tokens, Goyal et al 2024
>
> [9] A structural probe for finding syntax in word representations, Hewitt et al 2019
>
> [10] Scaling latent reasoning via looped language models, Zhu et al 2025
>
> [11] On the reasoning abilities of masked diffusion language models, Svete et al 2025

---

> > ### Comment · Reviewer_8ij9 · 2025-11-24
> > **Official Comment by Reviewer 8ij9**
> >
> > Thank you for your thorough response. You have answered all the questions I had about the work and addressed all of my main concerns.
> >
> > In light of the addition of an "Experimental Setup" section as well as the modifications to the proofs/clarifications of the theoretical approach, I will raise my score to an 8.

---

### Official Review · Reviewer_NBfP · 2025-11-01

**Soundness:** 3
**Presentation:** 3
**Contribution:** 3
**Rating:** 6
**Confidence:** 3

**Summary:**

This paper studies when looped transformers with padding tokens can recognize CFLs. The authors prove that looped transformers with $O(\log n)$ looping layers and $O(n^6)$ padding tokens are complete for all CFLs. for natural subclasses, the required padding can be reduced to $O(n^3).

**Strengths:**

The paper establishes a strong theoretical connection between CFL recognition and model size, which may provide useful insights for designing more efficient models.

**Weaknesses:**

1. The analysis focuses solely on model expressiveness and does not address learnability.

2. Although $O(n^3)$ padding is an improvement over $O(n^6)$, it may still be impractical in real-world settings.

3. The experimental section does not appear to relate clearly to the theoretical results, particularly concerning the number of padding tokens.

**Questions:**

See Weaknesses.

---

> ### Author Response · Authors · 2025-11-21
>
> Thank you very much for taking the time to read our work and writing a constructive review. We appreciate your insight of leveraging theoretical results on the expressivity of current models w.r.t. model size to develop novel architectures. In fact, we believe dynamically-scaling transformers are a first step towards developing more expressive architectures for inference-time computation. As an example, masked diffusion language models have recently been linked with padded- and looped-transformers ([2]).
>
> > The analysis focuses solely on model expressiveness and does not address learnability.
>
> We agree that theoretical learnability of CFLs is an interesting topic, but it is out of scope for our current research focusing on expressivity as a first step. Expressivity and learnability are distinct yet interlinked topics: when learning a formal language, a model might generalize reasonably well to some held-out data while the architecture’s inherent representational capacity prevents perfectly recognizing that language. For instance, if one trains a fixed-depth transformer on evaluating boolean formulas, we know that the model will never fully learn to solve this task unless it is augmented with looping layers, regardless of how the model is trained (as BFVP is log-depth complete). In other words, negative expressivity results imply negative learnability results.
> Therefore, our work centered around expressivity constitutes a first step on understanding the abilities and limitations of transformers on processing syntax, and hope future work on learnability can complete the picture.
>
> > Although n^3  padding is an improvement over n^6, it may still be impractical in real-world settings.
>
> We agree that n^3 may not be practical, though we do not view this as a weakness of our theoretical analysis. Our goal was to make progress analyzing the resources transformers require to recognize CFLs. The jump from n^6 to n^3 suggests recognition becomes more efficient for unambiguous CFLs, though we do not know if it is optimal. The theoretical machinery in our analysis provides a foundation for further theoretical and empirical exploration of whether transformers can do better than n^3 padding for recognizing unambiguous CFLs. As an example, we believe it might be possible to improve upon the n^3 space-bound on deterministic CFLs ([1]).
>
> Additionally, we note that n^3 padding does not require n^3 serial runtime because processing padding tokens (unlike CoT tokens) can be parallelized.
>
> > The experimental section does not appear to relate clearly to the theoretical results, particularly concerning the number of padding tokens.
>
> We agree with this remark, and acknowledge that padding experiments would have strengthened the paper. Nevertheless, our experiments still provide some new insights on the abilities of looped transformers in contrast to fixed-depth transformers: looping seems to improve performance when learning a log-depth complete language (namely, BFVP).
>
> [1] Time complexity of unambiguous path systems, Rytter 1982
>
> [2] On the reasoning abilities of masked diffusion language models, Svete et al 2025

---

> > ### Comment · Reviewer_NBfP · 2025-11-28
> >
> > I appreciate the clarifications in the rebuttal, which address my major concerns. I will maintain the positive score.

---

### Meta-Review · Area_Chair_ecvU · 2026-01-07

**Summary:**

Most of the reviewer concerns about clarity were answered, both theoretical and experimental. However, a recurring concern in the review process was a lack of clear value for the theory to affect any sort of practical usage across the board, from assumptions, to modeling choices, to having any experimental or interpretability connections, all of which was acknowledged by the reviewers as impractical. Secondarily, the crux of their theoretical proofs and bulk of experimental details has not been fully vetted, and so it seems wise for the paper to get a new set of reviews that can actually scrutinize the numerous details that were initially omitted as a very significant amount of details was added during the response period (including entire proofs!). The main strength is limited to this being somehow a valuable contribution towards transformer expressivity, yet none can verbalize how this is actually valuable. For these reasons I find this paper to remain at the borderline and would lean to reject as it is not a clear accept.

**Reviewer Concerns:**

Concerns addressed:
+ The theory as submitted was lacking in clarity. This resulting in a large number of clarification questions from reviewers, which included clarifying some quite significant components of the work  from defining crucial elements of the algorithm to replacing handwavy, informal proofs with formal arguments, in addition to a large number of detailed questions.
+ Key experimental details were not present in the main submission, which the authors added during the response period.

Concerns outstanding:
+ Value of the theory is questioned. One reviewer found the focus on expressiveness without learnability to be limiting. Multiple reviewers questioned the realism of the polynomial bounds and the specific modeling decisions (padding tokens, looping layers) as impractical in real world settings, which the authors more or less acknowledged as true. Another reviewer questioned the originality of the theory, which the authors gave a response about the literature being messy and differentiating between NC2 and AC1 that did not clearly answer the question.
+ Multiple reviewers felt the experiments to be inadequate. The experimental results do not directly relate to the theory, which authors acknowledge. In fact the authors further acknowledge that the theory is not tight and cannot predict what is learnable, which further questions the value.
+ Not all experimental details were provided. While an experimental details section was added, it occasionally omits details such as the actual length of OOD strengths (it only says they are longer than training) or what exactly a perturbation is of a positive string.
+ Reviewers felt interpretability results to be interesting here, which the authors agreed would be good to have but do not have. This is despite interpretability being marked as the primary area by the authors for this paper.
+ Correctness of the proofs have not been fully checked. This is because the initially submitted proofs were missing too many parts to be checked, with significant components being added during the response period without having been checked by a reviewer. While one reviewer’s clarity questions were answered (e.g. for the Lemmas), another reviewer had significant questions about the main, non-trivial part (Corollary 3.1). As this is the key part of the paper, it carries much weight in being validated.

**Reviewer Scores:**

Reviewer NBfP would have likely retained their score (6), as they gave a relatively light review without significant or detailed criticisms or strengths.

Reviewer 8ij9’s final score would have likely been the same (6). Their initial score may have updated as they stated they would do so if an experimental section was added. However upon a careful reading of the experimental section, this increase may have scaled back to their original score when the insufficiencies in the details were pointed out.

Reviewer YVFz likely would have maintained, or possibly lowered their score (4 or lower). It is unclear whether the main corollary is sufficiently detailed enough to justified, and the originality clarification was unclear.

Reviewer vZZD would have likely not changed their score (6), as the authors more or less acknowledge the main empirical weaknesses that were brought up.

---

### Decision · Program_Chairs · 2026-01-26

Reject